# Optimized Scenario for Estimating Suspended Sediment Yield Using an Artificial Neural Network Coupled with a Genetic Algorithm

Arvind Yadav [1], Mohammad Kamrul Hasan [2,*], Devendra Joshi [1], Vinod Kumar [1], Azana Hafizah Mohd Aman [2], Hesham Alhumyani [3], Mohammed S. Alzaidi [4] and Haripriya Mishra [5]

1   Department of CSE, Koneru Lakshmaiah Education Foundation, Vaddeswaram 522302, India
2   Faculty of Information Sciences and Technology, Universiti Kebangsaan Malaysia, Bangi 43600, Malaysia
3   Department of Computer Engineering, College of Computers and Information Technology, Taif University, Taif 21944, Saudi Arabia
4   Department of Electrical Engineering, College of Engineering, Taif University, Taif 21944, Saudi Arabia
5   Department of Civil Engineering, Gandhi Institute for Technology, Bhubaneswar 752101, India
*   Correspondence: mkhasan@ukm.edu.my

**Abstract:** Rivers are the agents on earth and act as the main pathways for transporting the continental weathered materials into the sea. The estimation of suspended sediment yield (SSY) is important in the design, planning and management of water resources. The SSY depends on many factors and their interrelationships, which are very nonlinear and complex. The traditional approaches are unable to solve these complex nonlear processes of SSY. Thus, the development of a reliable and accurate model for estimating the SSY is essential. The goal of this research was to develop a single hybrid artificial intelligence model, which is a hybridization of the artificial neural network (ANN) and genetic algorithm (GA) (ANN-GA) for the estimation of SSY in the Mahanadi River (MR), India, by combining data from 11-gauge stations into a single hybrid generalized model and applying it to every gauging station for estimating the SSY. All parameters of the ANN model were optimized automatically and simultaneously using GA to estimate the SSY. The proposed model was developed considering the temporal monthly hydro-climatic data, such as temperature (T), rainfall (RF), water discharge (Q) and SSY and spatial data, including the rock type (RT), catchment area (CA) and relief (R), of all 11 gauging stations in the MR. The performances of the conventional sediment rating curve (SRC), ANN and multiple linear regression (MLR) were compared with the hybrid ANN-GA model. It was noticed that the ANN-GA model provided with greatest coefficient of correlation (0.8710) and lowest root mean square error (0.0088) values among all comparative SRC, ANN and MLR. Thus, the proposed ANN-GA is most appropriate model compared to other examined models for estimating SSY in the MR Basin, India, particularly at the Tikarapara measuring station. If no measures of SSY are available in the MR, then the modelling approach could be used to estimate SSY at ungauged or gauge stations in the MR Basin.

**Keywords:** Mahanadi River; genetic algorithm; suspended sediment yield; artificial neural network; water discharge

## 1. Introduction

The transport of sediments by rivers to the oceans is a vital link between the terrestrial and marine ecosystems [1,2]. The quantity of suspended sediment yield (SSY) in rivers is always an important element when evaluating dam filling, flooding risk, reservoir sedimentation, hydropower equipment lifetime, aquatic ecosystems, changes in nutrient cycling, irrigation schemes and increasing the cost of water treatment [3–8]. Moreover, the deposition of SSY diminishes the water storage capacity, and high SSY in the river's downstream section causes channel migration on the lateral basic. The channel migration



causes significant flooding during heavy rains during the monsoon season [8,9]. Various researchers have studied flooding due to high SSY and its harmful impacts on land use, soil fertility, flood plain, loss of human life and property losses [8,10,11]. Moreover, various researchers have revealed the impact of the suspended sediment on the erosion of hydro-turbine components in hydro power plant, which is one of the most challenging problems [7,12–14]. The prediction of the quantity of sediment that will be present in a river at a given time helps planners and managers of water resource systems to better understand the system in terms of its problems and to find alternative ways to address those [15]. Thus, the measurement of SSY is becoming essential, but its estimation is difficult because it is affected by various controlling factors and their interlationsships, which are highly nonlinear and complex [16–19].

River sediment and continental erosion are sensitive to many factors, such as water discharge (Q), runoff, temperature (T), geomorphology, basin geology, relief, soil types, basin area and channel slope [20–25]. Sediment yield and water discharge have a strong nonlinear association in many of the world's river systems [26–31]. Climatic factors, such as humidity, temperature, wind, solar radiation and precipitation, play an important role in sediment generation and river transportation [32–35]. Jansson [36] investigated the effect of river system factors, such as geology, soil and rainfall (RF), on SSY. However, SSY is highly dependent on RF intensity. As a peninsular river, the majority of the water discharge in the Mahanadi River is contributed by precipitation during the monsoon season, and groundwater recharge accounts for a small contribution [29,31]. Rainfall is an important factor, which, in many ways, affects the groundwater resources in an area. There is also a considerable flow of water in some locations during the non-monsoon season, which may be due to additions from groundwater to the river during non-monsoon periods [29,31]. Sedimentation is caused by the effects of rainfall splash detachment and entrainment through overland flow. Temperature changes may influence sediment discharge by altering runoff and changing the rate of erosion due to their effects on evapotranspiration, vegetation and weathering [25,37]. The most dominant factor for mechanical denudation in a basin is basin relief, where steep catchments are associated with a high rate of erosion and sediment load. One of the major factors influencing soil particle detachment and transport is slope gradient [38]. The variation in catchment properties, such as gradient and storage capacity, influences sediment yield [24]. The type of rock is also an important controlling factor for erosion. In this study, temporal data (water discharge (Q), rainfall (RF) and temperature (T)) along with spatial data (rock type (RT), relief (R), and catchment area (CA)), as major controlling factors of the SSY, were used to develop the various SSY prediction models on the basis of previous research [20–25,38].

Traditional mathematical models, such as multiple linear regression (MLR) and sediment rating curve (SRC) for SSY prediction, have been adopted in previous studies but were unable to capture the complex, erratic variations in SSY [17,37,39,40]. The MLR models can capture any linear relation, but these are incapable of modelling the existence of highly nonlinear in SSY. The SRC model's main limitation is that it can only take into account a single independent factor (Q), and nonlinearity only follows the power law function, i.e, the power relation model for estimating the SSY. However, many studies have shown that multiple factors, such as T, RT, R, CA, Q and RF, significantly impact SSY [20,21,24,41–43]. Thus, the traditional SRC method is not suitable for the prediction of SSY.

On the other hand, artificial intelligence (AI)-based techniques are capable of estimating complex nonlinear phenomena and are now widely used by various researchers in different scientific domains [30,40,44–52]. The artificial neural network (ANN) is a very popular artificial intelligence-based technique which is widely applied successfully in interdisciplinary domains [53–55]. Numerous researchers have notably used several ANN algorithms to estimate and forecast the suspended sediment yield [17,30,40].

The major drawback of artificial intelligence-based techniques, such as ANN, is overfitting and underfitting problems due to inappropriate methods of selecting the ANN parameters using grid-searching or trial-and-error procedures [40,56,57]. These techniques

may not guarantee the optimum parameter and topology selection in ANN models. Moreover, the trial-and-error methods are a time-consuming process. Therefore, the selections of learning parameters and network topology in ANN models are the essential task to develop efficient and robust ANN models for predicting the SSY. The genetic algorithm (GA) is an optimization algorithm based on population and the principle of Darwin's evolution, which is used to determine the optimum ANN's parameters [58]. It uses genetic operators, including crossover, selection, and mutation, to generate variations in a chromosome of a problem statement. The GA is a popular optimization method for solving nondifferentiable, noncontinuous, stochastic or complex nonlinear problems in noisy environments [59,60]. It is one of the most popular global search optimization algorithms which is used in several applications [19,40,61–64]. The artificial intelligence may have some drawbacks in dealing with nonstationary data due to the poor selection of model parameters [40,61–63]. The hybrid modeling approaches, which include distinct data preprocessing and combine techniques, was proposed to enhance the generalization capability of individual artificial intelligence-based methods. Numerous studies have been conducted recently in a variety of fields using hybridized genetic algorithm-based artificial neural networks, which are gaining popularity among researchers. These studies have shown that these hybrid approaches are capable of enhancing the system's accuracy. The hybrid artificial intelligence has demonstrated the use of heuristic and meta-heuristic approaches for simultaneous optimization of associated parameters in artificial intelligence models to overcome the limitation of trial-and-error methods and overfitting and underfitting problems. Several studies have shown that these approaches not only reduce the computational intensiveness but also provide superior results. The overfitting and underfitting problems of ANN are overcome by hybridization of ANN with GA [40,61,65].

Few hybrid artificial intelligence techniques have been used recently for modeling with multiple model parameters optimization [40,64,66–68]. Several studies have shonw that GA-based ANN models outperformed traditional ANN and regression models in terms of prediction accuracy [40,61,69,70]. The GA-based ANN models have been fruitfully applied from the prediction, as well as the forecasting perspective, of stream flow, flood, bed load transport, Q and run-off [69,71–74]. Various researchers have applied GA and GA-based ANN models for prediction in hydrology and other domains [70,75–80]. Sirdari et al. [74] employed a genetic programming (GP)-based ANN hybrid approach for estimating the bed load transport of the Kurau River in Malaysia. The GA, ANN and GA-based ANN approaches have been successfully applied in other settings for predicting sediment in river basin systems [40,74–77].

In terms of flood-producing capacity, water potential and SSY, the Mahanadi River is the second biggest Indian peninsular river [31,81]. Few researchers have applied the ANN and GA algorithm for predicting suspended sediment at specific single gauging stations in Mahanadi River (MR) using temporal data only [30,82,83]. In that study, all model parameters were selected using trial-and-error approaches. Spatial data, such as rock type, relief and catchment, were also not considered as controlling factors of SSY in the model development, but these are the major factors of the SSY. Furthermore, no attempt has been made to development of fully automatic highly generalized global single hybrid artificial intelligence based prediction model using huge amount of temporal and spatial data of various gauging stations for predict the SSY and applying this single model instead of various model at individual gauging stations for estimating the SSY in the MR Basin. In this research, a single hybrid ANN coupled with GA (ANN-GA) model was developed using a large amount of combined temporal data (Q, RF, T and SSY) and spatial data (RT, R and CA) from 11 gauging stations for efficiently estimating the SSY at individual gauge stations among all 11 gauging stations in the MR. All ANN model parameters were optimized simultaneously using the GA, which overcame the drawback of traditional trial-and-error approaches. The efficiency of the hybrid ANN-GA was compared to the ANN, MLR and SRC methods for evaluating the estimation capability of models. The hybrid SSY estimation model optimized multiple model parameters simultaneously using temporal variables

(Q, RF and T) and spatial variables (RT, R and CA) as inputs. The research's key contribution is the development of a completely autonomous, highly generalized global hybrid artificial intelligence model with optimized ANN multiple model parameters simultaneously using single-objective GA to estimate SSY where only minimal human involvement is needed. There is no such study available which discusses the simultaneous optimization problem of all ANN training parameters, including inputs and network topology together, for SSY estimation in a river basin. Furthermore, the MR study provides some insight into the use of artificial intelligence in SSY estimation in the MR system by combining data from 11 gauging stations into a single generalized model and applying it to each gauging station. In terms of the study area, data employed and outcomes gained, this research is unique for SSY estimations. To the best of our knowledge of author, there is no such study available for suspended sediment yield estimation in the river basin. We adopted the technique of using genetic algorithm as a meta-heuristic algorithm to optimize all parameters of the artificial intelligence model (artificial neural network)(inputs, transfer functions, number of hidden neurons, combinational coefficient, and initial network weights and bias terms) for sediment yield prediction in the MR basin system. Our methodology may not be unique, but its application in sediment load prediction is unique. Moreover, the case study of the Mahanadi River provides some insight into the application of artificial intelligence method in sediment prediction in river system. The proposed ANN-GA model outperformed traditional the MLR, ANN and SRC methods in the testing phase. The results indicated that the hybrid ANN-GA-GA model performed well and delivered a higher performances and better SSY prediction capability than the traditional models. Overall, the proposed hybrid ANN-GA artificial intelligence approaches are recommended for the prediction of SSY in MR River because of their reasonably better performance and ease of implementation.

## 2. Study Area

The Mahanadi River was selected for the study of suspended sediment yield estimation. It is the fourth largest Indian river basin, with 141,589 km$^2$ area in total catchment [84]. The longitude and latitude range from $80°30'$ to $86°50'$ east and from $19°20'$ to $23°35'$ north, respectively. It flows through the states of Maharashtra, Jharkhand, Chhattisgarh and Odisha. The catchment area contribution of the river is 53% (75,136 km$^2$) in Chhattisgarh and 46% (65,580 km$^2$) in Odisha, while with remainder of the basin is in the Maharashtra and Jharkhand states [84]. The total length of the river is 851 km when entering the Bay of Bengal. In 2005–2006, agricultural land covered the majority (54.27%) of the Mahanadi River Basin, followed by forest cover (32.74%), wasteland (5.24%), water bodies (4.45%) and build-up land (3.30%) [81]. The Mahanadi Basin has a surface elevation from 30 to 700 m from the mean sea level [85]. The largest earthen dam in world, i.e., Hirakud Dam and Chilka Lake, are two large water bodies present in the MR basin. The basin's gauge height ranges from 50 to 411 m, according to the Central Water Commission (CWC) [86]. The elevation of the basin is shown in the Figure 1, which shows the mainstream of the Mahanadi River Basin and geographical locations of all eleven gauging stations. The maximum drainage area of 124,450 km$^2$ is covered by Tikarapara (farthest downstream station), while Andhiyarakhore (upstream station) covers a minimum drainage area of 1100 km$^2$ in the MR Basin.

The average annual rainfall in the entire MR Basin varied from 1200–1400 mm for the period 1971–2004 [81]. The MR Basin receives nearly 90% of the annual rainfall during monsoon period (from June to October). The coldest months of the year are December and January, having the lowest temperature of 12 °C, whereas April and May register the highest temperature, ranging from 39 °C to 40 °C for the period 1969–2004 on the basis of daily data [81,87].

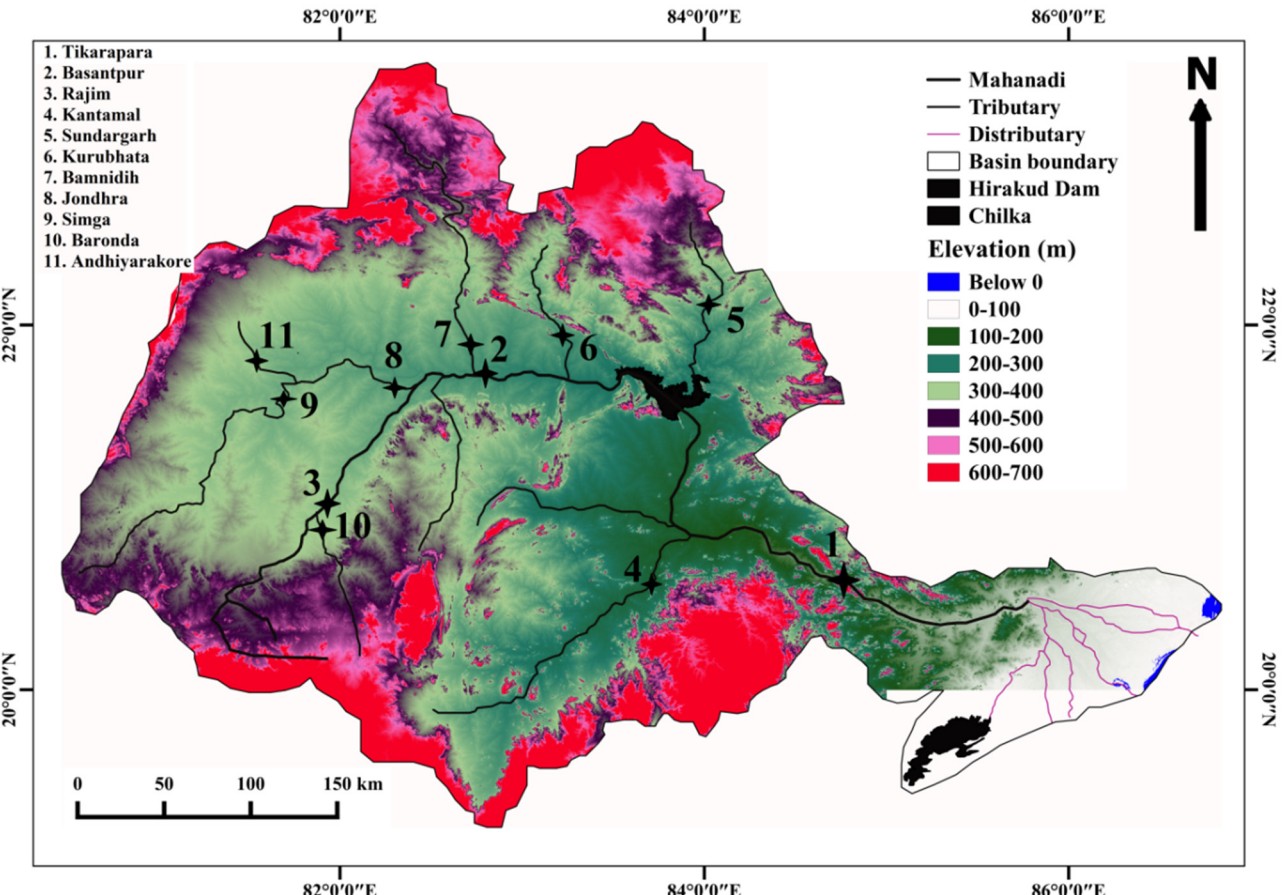

**Figure 1.** Elevation map of the Mahanadi Basin showing the main streams and locations of gauging station [30,77].

Proterozoic sedimentary rocks are generally found at the upstream of the river, which occupies an area of 57,780 km$^2$, whereas the downstream parts of the river have silicate rocks of metamorphic origin [88]. The basin area's lithologies are composed of 34% granite suite, 7% Khondalite suite, 15% charnockite suite, 17% limestone, 22% sandstone and shale and 5% coastal alluvium [89].

## 3. Methodology and Data Used Description

The normalization and division of the data were the pre-processing steps before ANN-GA modeling for estimation of SSY. Normalization is a technique for removing disparities in the dimensions and ranges of data. All variables were fixed within the range of 0 to 1 to perform normalization, where 0 is the minimum value and 1 is the maximum value [30,40]. Data normalisation speeds up data computation and convergence throughout training [90].

In this study, all gauge stations contains temporal monthly data, such as RF, T, Q and SSY from 1990 to 2010, as well as spatial data, including R, CA and RT, with the exception of the Kantamal gauge station, which has temporal data from 1990 to 2008 in the MR for developing the ANN-GA models. All data were collected from the Central Water Commission (CWC), Mahanadi Bhawan, Bhubaneswar, Odisha. Data were divided into training, validation and testing data. Training data (70 percent of the dataset) were used to construct the models. Testing data (15 percent) were utilized in the proposed model to assess the model's performance. Validation data (15 percent) were used to avoid overfitting of the developed models. Testing data are unseen data that are not used in the model development process [30,40,65,77,91]. The training data were taken from 1 June 1990 to 31 May 2004, and validation data were taken from 1 June 2004 to 31 May 2007. Similarly, for testing purposes, data were listed from 1 June 2007 to 31 May 2010. In addition, at the

Kantamal gauge station, the validation data ranged from 1 April 2002 to 31 December 2005, training data ranged from 1 June 1990 to 31 March 2002 and testing data ranged from 1 January 2005, to 30 September 2008. Finally, all gauge stations' data were integrated to form a single MR training, validation and test data set.

The GA-ANN model's customized code was developed in Matlab 2021 software, and ANN parameters were chosen using the normalized data. The Levenberg–Marquardt (LM) back-propagation algorithm was employed in this study to predict the SSY using multi-layer perceptron based ANN with such a computational efficiency training process [40,85]. The details description about multi-layer perceptron and Levenberg–Marquardt training algorithm have been discussed by various researchers [30,40,92]. The inputs, activation function, neurons in the hidden layer and the initial weights values affect the performance of multi-layer perceptron based ANN. The incorrect selection of any of these factors may lead to a poor ANN model. There are many drawbacks of traditional grid-searching or trial-and-error and methods of the ANN. To overcome the limitation of these approaches, simultaneous multiple parameter optimization using GA helps to reduce the time in computational and deliver an efficient result [40,66,70]. In this research, an ANN model was developed, where all parameters of the ANN (network initial weights, hidden layer nodes, inputs, transfer function and combination coefficient of Levenberg–Marquardt) were optimized simultaneously using GA. These selected optimum parameters were applied for SSY for estimation in the MR using temporal data (RF, T and Q) along with spatial data (RT, R and CA) as inputs.

The GA is considered as global optimization algorithm which is used for selecting the optimum model parameters. In the GA, various genetic operators are included, such as mutation, selection and crossover, which are used to identify variety in populations of individuals with specific problems. The GA was combined with a multi-layered feed-forward ANN. The Levenberg–Marquardt algorithm was used to train the ANN. The GA was used for simultaneous optimization of all parameters of the ANN model. The parameter selection and ANN training were started at the same time to produce a more robust solution with a lower possibility of trapping in a local optimal point. The purpose of using GA in this study was to produce successive populations by selecting 5 major parameters (inputs, combination coefficient, transfer function, number of neurons and bias and connection weights) of ANN. Each of the five ANN parameters was assigned to a binary sequence known as a chromosome. Numerous chromosomes were activated at random and upgraded repetitively by genetic operations such as crossover, selection and mutation to produce a better solution. Each chromosome was divided into 5 parts, each of which represents a different ANN parameter. The chromosome's primary parts represent the inputs, while the secondary parts represent the transfer function of the ANN. In this study, 3 different transfer functions, i.e., log-sigmoidal, tan-sigmoidal and linear, were tested. The third part of the chromosome contained the hidden layer node count. The maximum number of hidden neurons was limited to 32 due to the complexity of the model and the heavy computational time. The fourth part of chromosome represents the combination coefficient ($\mu$). The fifth part represents the bias and connection weights of ANN models.

The preferred pairing of parent's chromosomal were taken from the initial population, and offspring were created in succeeding generations based on the best-fitting chromosomes. The initialization of the chromosome is uniform. The chromosome number in the population was set to 50 for reducing the processing time and maintain diversity. Training data were trained for each chromosome using the ANN model, and the validation data calculated the fitness value. The next generation would choose the best individual chromosome with the best fitness function. Elitism selects some chromosomes depending on individual fitness values. The mutation and crossover operations are carried out based on the computed fitness value [93]. The mutation operation helps the algorithm to escape the solution from local minima. The low constant mutation probability was used in this study as 0.05. In this study, a uniform crossover with probability rate 0.6 was used. The

roulette-wheel selection approaches was used in selecting the elite members of a population. Finally, reproduction occurred through recombination to produce offspring. The number of elites moved to next generation of GA was two [94]. The crossover operation was used to find a better solution. New individual chromosomes benefitted, and these benefits were obtained from the parent fitness through the crossover operation. Mutation were used and were responsible for population diversity.

The GA's success rate is determined by a high likelihood of crossover and a low likelihood of mutation [95]. A uniform crossover was explored with a probability rate of 0.6 [62]. The algorithm can move away from local minima with the help of the mutation operation. To keep the algorithm from going into a random search, the fixed mutation probability was set to 0.05. The fitness function for GA was the root mean square error (RMSE) of the training phase. The RMSE fitness function was used to assess the fitness values of all chromosomes. To keep the population size at 50, chromosomes with low fitness function values were deleted after each generation. The resulting population of chromosomes became the beginning solution for the next generation after one generation. The genetic process was carried out until the terminating requirement was reached, and the least RMSE of the fitness value was obtained. In this research, the greatest number of generations was 50, which was used as stopping criteria [70,96]. The genetic operation was performed until reached the stopping criterion like maximum generation. The best solution was obtained at the final generation when the minimum RMSE of fitness value was reached. The population chromosome associated with the best solution involved the best learning parameters (transfer functions, number of hidden neurons, µ value, and initial network weights and bias terms) for the ANN model.

The SRC and MLR regression methods were used for comparison to check the predictive capability of models. The SRC, also known as the power relation model, is a popular method for maintaining nonlinear relationships between output and input variables. It calculates the amount of SSY corresponding to the Q [16]. The traditional MLR is a statistical approach for maintaining the linear relationship between the various input (independent) variables by modifying the linear equation to data and predicting the output (SSY).

The most common type of automatic water level recorder uses a float line with a metal float at one end and small counterweight at the other end. The float line passes over a pulley and transfers the changes of water level to it. A recording stylus is attached to the pulley. It moves laterally and traces the water level fluctuations on a recorder chart. The recorder chart is a tracing quality strip paper wound over rollers or a drum. The recorder chart is connected to a clockwork mechanism, which moves it continuously at predetermined speed. Suspended sediment concentrations are typically measured by collecting samples of water-sediment mixtures. Bottle samples are the traditional method for obtaining suspended sediment samples and may be collected using either depth-integrated or point-integrated methods [97,98]. The depth-integrated sampling method is generally used, which involves lowering the sediment samples from the river surface to the bed of the channel at a uniform rate while a bottle within the sampler collects an incremental volume of the water-sediment mixture from all points along the sampled depth. Each location chosen for a measurement is known as a sampling vertical and the movement of the sampler from the surface to the bed, or vice versa, is known as a transit.

## 4. Results and Discussion

### 4.1. T-Test of Data and Spatial Variation of Data

The *t*-test was performed to check the similarity of distribution of the training, validation and testing data types. The results of the paired sample the *t*-test of training, validation, and test data are presented in Table 1. It can be seen that $p$ values, i.e., the probability of acceptance of a null hypothesis of the *t*-test, were greater than 0.05 for all paired sample tests. Therefore, the null hypothesis of *t*-test was accepted at the 5% confidence level. It shows that the training, validation and testing data are statistically similar in nature.

**Table 1.** *T*-test values of the hydro-climatic data set.

| Data Set | *t*-Test | Water-Discharge | Rainfall | Temperature | Suspended Sediment Yield |
|---|---|---|---|---|---|
| Training and testing | *p* | 0.9013 | 0.5932 | 0.2339 | 0.2392 |
|  | t | 0.1230 | 0.5328 | 1.1632 | −131.2 |
| Training and validation | *p* | 0.9630 | 0.8957 | 0.2863 | 0.0855 |
|  | t | −0.0352 | −0.1311 | 1.0663 | 1.7205 |
| Validation and testing | *p* | 0.8838 | 0.5935 | 0.9259 | 0.5603 |
|  | t | 0.1339 | 0.5326 | 0.0930 | −0.5827 |

The histogram plots of monthly hydro-climatic (Q, R, T and SSY) data are presented in Figure 2. It is observed that the water discharge, rainfall and suspended sediment yield were positively skewed (right asymmetry), while the temperature was negatively skewed (left asymmetry). Suspended sediment yield had a higher skew than other variables. High skewness values have a negative impact on performance of the ANN model [99]. It was observed that the monthly average T varied in the basin from 14 °C to 39.5 °C. The highest and lowest T were found at Basantpur and Kantamal, respectively. Variation in rainfall was found to range from 0 to 1222.7 mm in the MR Basin. The highest rainfall was found at Tikarapara. Te maximum monthly Q was 430,767 cumec(m$^3$/s) at Tikarapara. The SSY in the basin varied from 0 to 17,346,901 tons/month. The maximum SSY was found at Tikarapara.

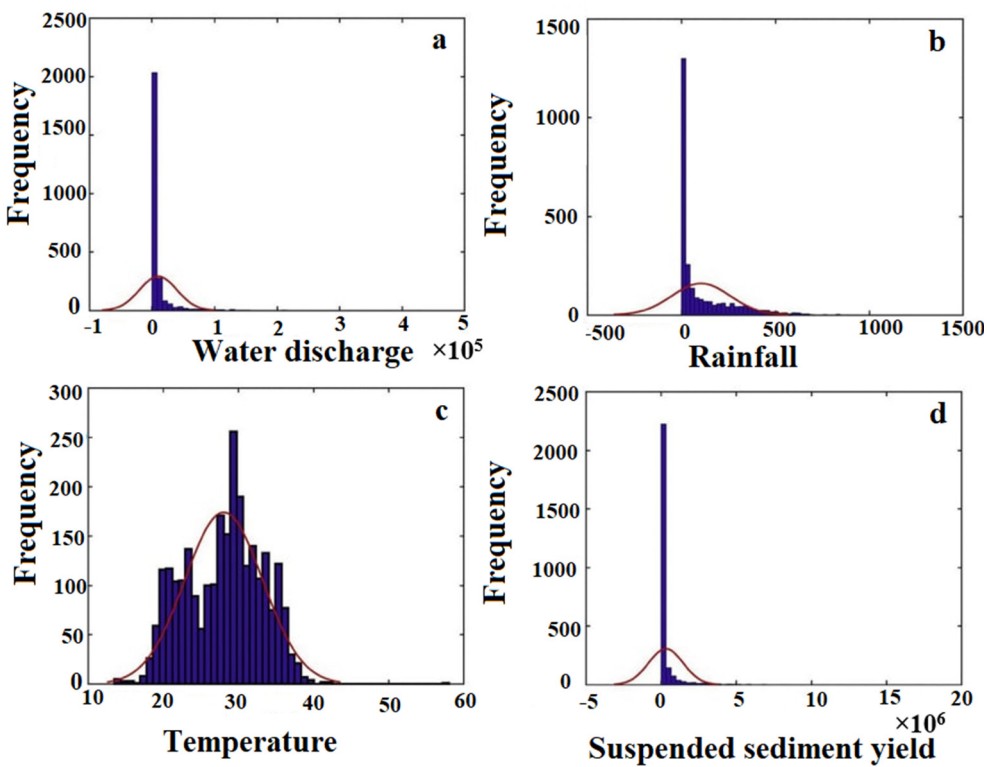

**Figure 2.** Histogram plots of hydro-climatic data in Mahanadi River Basin: (**a**) water discharge, (**b**) rainfall, (**c**) temperature, (**d**) suspended sediment yield.

The spatial variation of hydroclimatic data is given in Figure 3. The distributions of Q at different locations are shown in Figure 3a. The MR's tributaries had a wider variety of Q, ranging from 4.055 km$^3$/year at Andhiyakore to 588.1 km$^3$/year at Tikarapara. The mainstream stations, such as Tikarapara (588,119 m$^3$/year) and Basantpur (248,384 m$^3$/year), and major tributaries, including the Seonath at Jhondhra (97,569.83 m$^3$/year) and the Tel at Kantamal (145,137 m$^3$/year), showed relatively larger values of Q. Baronda (18,122 m$^3$/year), Simga (58,392 m$^3$/year), Mand at Kurubhata (27,593 m$^3$/year), Ib at Sundargarh (38,978.4 m$^3$/year), Rajim (365,56.1 m$^3$/year) and Bamnidih (48,781 m$^3$/year) showed

relatively lower values of Q. In the downstream of the Mahanadi River, water discharge increases due to confluence of various tributaries in the basin. The peninsular MR receives most of the water in monsoon season through RF. It shows a more seasonal variation as it is supported from monthly RF, which is controlled by strong monsoons. The maximum monthly Q of 430,767 $m^3$/second was found in July 1994, which was the maximum average monthly Q in the basin from 1990 to 2010. The minimum monthly Q at Andhiyarakore was zero cummec appeared in May 2010.

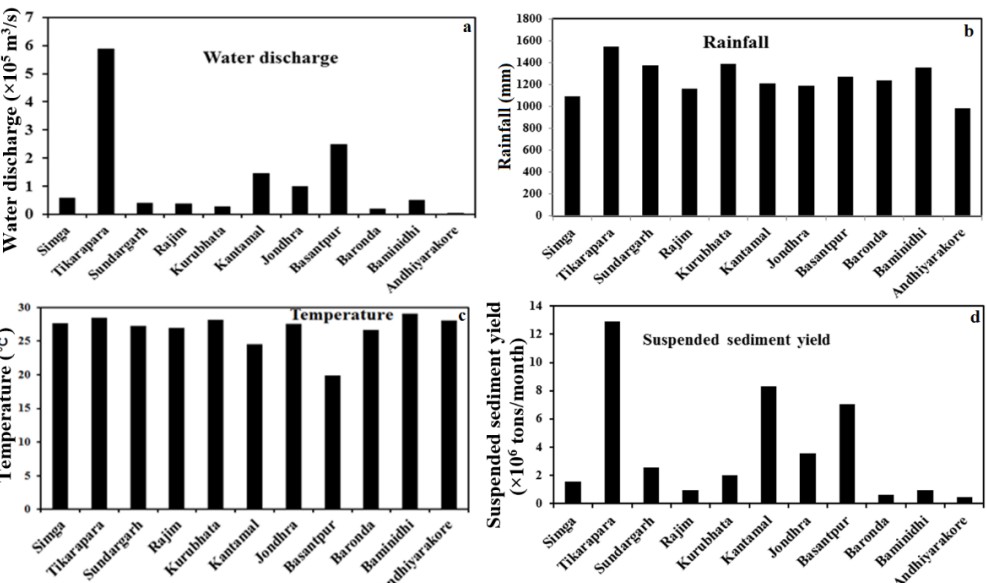

**Figure 3.** Average annual variation of hydroclimatc data at different stations of the MR Basin: (**a**) water discharge, (**b**) rainfall, (**c**) temperature, (**d**) suspended sediment yield.

In the Table 2, r1, r2 and r3 represent the linear correlation of Q and SSY, RF and SSY and, T and SSY, respectively. It is observed from the Table 2 that SSY had relatively lowest value of linear correlation with the Q at the Bamnidih gauging station among all gauging stations. This may be due to the large Bango Dam near this gauging station. Highest linear correlation was found between the SSY and Q at Tikarapara among all gauging stations. This may be attributed to the highest SSY, CA, Q and RF values at Tikarapara, which is situated in the farthest downstream station of the MR Basin before the Bay of Bangal. The r1 value of the Pearson correlation coefficient between the SSY and Q is high and significant. Thus, water discharge has more a significant contribution on SSY. The r2 value of the Pearson correlation coefficient between the rainfall and suspended sediment yield is significant. It indicates that there was a significant contribution of rainfall on suspended sediment yield. It is also clear from the Table 2 that r3 values of the Pearson correlation coefficient between the temperature and suspended sediment yield is low and not significant. It means that temperature had no direct significant contribution on suspended sediment yield but affects the suspended sediment in indirect ways.

In the Mahanadi Basin, rainfall (RF) is distributed spatially, as shown in Figure 3b. Rainfall is a key source of water for the river basin. The minimum and maximum averages annual RF values in the basin were 982 mm at Andhiyarakore and 1549 mm at Tikarapara, respectively, during 1990–2010. Maximum annual average RF of 2456 mm/year from June 2001 to May 2002 over 20 years (1990–2010) was recorded at Tikarapara. A maximum monthly average RF of 939 mm at Tikarapara station was found in July 2001. Panda et al. [100] demonstrated that the average annual RF in the MR is higher in comparison to the majority of Indian tropical rivers. Among all gauging stations, Tikarapara (1549 mm) showed maximum annual average RF, followed by Kurubhata (1388 mm), Sundargarh (1378 mm), Bamnidih (1356 mm), Basantpur (1270 mm), Baronda (1239 mm), Kantamal (1210 mm), Jhondhra (1188 mm), Rajim (1163 mm), Simga (1091 mm) and Andhiarakhore (981 mm).

Panda et al. (2013) also found similar temporal and spatial variations in RF distribution. The distribution of RF was uneven in the MR Basin [101].

**Table 2.** Pearson correlation coefficient (r) of the hydro-climatic data.

| Stations | Q-SSY(r1) | RF-SSY(r2) | T-SSY(r3) |
|---|---|---|---|
| Tikarapara | 0.9323 | 0.5787 | 0.1499 |
| Simga | 0.8528 | 0.5736 | −0.0865 |
| Andhiyarakhore | 0.8218 | 0.5847 | 0.1866 |
| Sundargarh | 0.8913 | 0.7917 | 0.1459 |
| Bamnidih | 0.7924 | 0.4963 | 0.1082 |
| Jondhara | 0.8873 | 0.5711 | 0.1437 |
| Kantamal | 0.8492 | 0.6643 | 0.1038 |
| Kurubhata | 0.9031 | 0.7866 | 0.1734 |
| Basantpur | 0.8935 | 0.6941 | 0.1516 |
| Baronda | 0.8224 | 0.6467 | 0.0677 |
| Rajim | 0.8413 | 0.6377 | 0.0624 |

Variations in annual average T at various gauge stations of MR basin are shown in Figure 3c. Annual average T ranged from low as 20 °C at Basantpur to high as 29.5 °C at Bamnidih in the basin. December or January is the region's coldest month, whereas April or May is this hottest month. Maximum mean monthly T in the basin was found to be 39.5 at Kantamal in May 2005. Minimum mean monthly T was 14 °C at Sundargarh in January 2010. The MR basin achieved the lowest and highest temperatures during the winter and summer season, respectively [31,87]. T indirectly affects RF distribution [102–107].

Annual average SSY varied from 458,364 tons/year to 12,940,610 tons/year based on 20 years of data from 1990 to 2010 at different places in the MR, which are shown in Figure 3d. The mean annual SSY values at the main stream stations, such as Tikarapara and Basantpur, and major tributaries, including the Seonath, the Mand, Ib and Tel, showed a relatively higher value of SSY. Baronda, Simga, Rajim and Bamnidih showed relatively lower values of SSY. The annual SSY at Tikarapara varied from 2,170,793 tons/year (2002−2003) to 50,265,601 tons/year (1994–1995). Monthly maximum SSY was 17,346,901 tons/month at Tikarapara in July 1994. The lowest annual average SSY was found at Andhiyarakhore. The annual SSY at Andhiyarakore varied from 20,796 tons/year (2009−2010) to 1,764,906 tons/year (1994–1995). The maximum SSY was 711,518 tons/month at Andhiyarakore in October 1994. During the monsoon season, when RF was higher, the SSY production was higher.

*4.2. Hybrid ANN-GA Model for Estimation of SSY*

The proposed hybrid ANN-GA model provides a list of optimal solutions based on the maximum generation threshold criteria value. During the training stage, Figure 4 represents the change in mean fitness and best fitness values in each generation. The best fitness had a score of 0.00211, while the average fitness score was 0.00707. The best fitness function of each genetic learning generation remained constant after 31 generations (Figure 4). The results also indicate that the optimum hidden layer neurons was 32. The pure linear and tan sigmoidal activation functions are optimally chosen for the output and hidden layers, respectively. The ANN-GA model selected 189 as the initial optimized value of μ in the Levenberg–Marquardt algorithm. The initial bias and connection weights were optimally chosen, and the total number of terms in this case was 257. The parameters selected by the ANN model using the GA were the optimum solution.

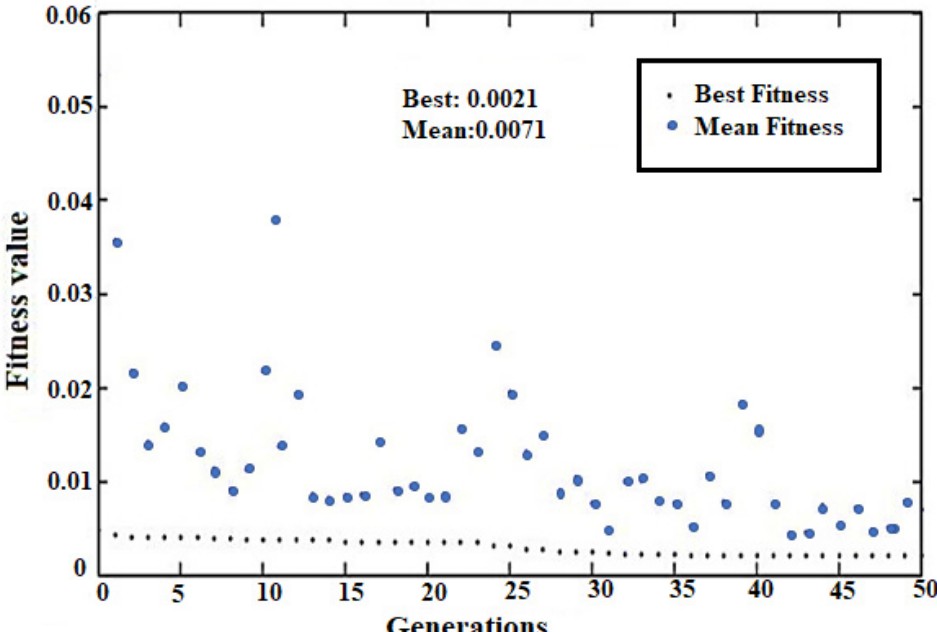

**Figure 4.** The fitness function profile of each generation during ANN-based GA learning.

The ANN-GA evaluation was performed using the root mean square error (RMSE), mean absolute error (MAE), mean square error (MSE), mean error (ME), variance of error(VAR), coefficient of correlation (r) and coefficient of efficiency (CE) statistical measures. The testing data were used to evaluate the proposed ANN-GA model's generalization ability and performance. Table 3 shows the statistical analysis of errors based on the actual and model simulated SSY for the testing, training and validation data. The results indicate that during the testing, training and validation phases, the RMSE (0.0059–0.0119) and MAE (0.0023–0.0035) values were lower, but the r was quite higher (0.7519–0.9721). Based on these error statistics, it can be inferred that such a ANN-GA model predicts SSY with reasonable accuracy. The generated model's ability to generalize was shown by the uniformity of all variables across all three different data sets. It was demonstrated that the ANN-GA model may prevent over- and under-fitting by employing low error values and higher r values in the testing, training and validation phase. Due to similar goodness-of-fit criteria values, the performances in training, validation and testing had similar qualities, as shown by the additional statistics in Table 3. The best-fitting model findings demonstrate that the Levenberg–Marquardt algorithm, combined with GA, improved the generalisation of the ANN. It is observed from Table 3 that Andhiyarakore had the lowest value of r (0.687) and a moderate correlation between the predicted and observed SSY. The ANN-GA model cannot provide satisfactory performance at this gauging station. The Tikarapara gauging station showed the highest r value (0.980).

The coefficient of efficiency value ranged between −258.929 and 0.9355. Andhiyarakhore, Kurubhata, Rajim and Bamnidih had negative coefficient of efficiency values, indicating that the ANN-GA performance was poorer than the observed mean value. The coefficient of efficiency value was found to be 0.9355 at Tikarapara, which is almost close to 1, showing the greatest value among 11 stations. Thus, the ANN-GA model exhibited the best prediction model at Tikarapara. Moreover, the coefficient of efficiency values varied from 0.2513 to 0.772 for the remaining gauging stations, which indicates good accuracy in estimating the SSY.

**Table 3.** Error statistics of the ANN-GA model in the testing, training and validation phase.

| ANN-GA | RMSE | MSE | MAE | Variance | r | Coefficient of Eficiency |
|---|---|---|---|---|---|---|
| Training | 0.0048 | $2.390 \times 10^{-05}$ | 0.002 | $2.390 \times 10^{-05}$ | 0.972 | 0.956 |
| Validation | 0.014 | $2.000 \times 10^{-04}$ | 0.004 | $1.000 \times 10^{-03}$ | 0.752 | $-0.081$ |
| Testing | 0.009 | $7.660 \times 10^{-05}$ | 0.003 | $7.550 \times 10^{-05}$ | 0.871 | 0.667 |
| Tikarapara | 0.007 | $5.260 \times 10^{-05}$ | 0.006 | $5.530 \times 10^{-05}$ | 0.980 | 0.936 |
| Simga | 0.008 | $5.890 \times 10^{-05}$ | 0.001 | $5.820 \times 10^{-06}$ | 0.921 | 0.251 |
| Andhiyakore | 0.001 | $8.550 \times 10^{-07}$ | 0.001 | $2.830 \times 10^{-07}$ | 0.688 | $-18.810$ |
| Sundargarh | 0.004 | $1.710 \times 10^{-05}$ | 0.002 | $1.750 \times 10^{-05}$ | 0.721 | 0.558 |
| Bamnidih | 0.005 | $2.090 \times 10^{-05}$ | 0.003 | $1.900 \times 10^{-05}$ | 0.906 | $-259$ |
| Jondhara | 0.005 | $2.950 \times 10^{-05}$ | 0.003 | $2.920 \times 10^{-05}$ | 0.830 | 0.627 |
| Kantamal | 0.032 | $1.000 \times 10^{-03}$ | 0.013 | $7.000 \times 10^{-04}$ | 0.778 | 0.259 |
| Kurubhata | 0.004 | $1.900 \times 10^{-05}$ | 0.003 | $1.630 \times 10^{-05}$ | 0.732 | $-2.201$ |
| Basantpur | 0.007 | $5.120 \times 10^{-05}$ | 0.005 | $3.970 \times 10^{-05}$ | 0.917 | 0.772 |
| Baronda | 0.001 | $1.770 \times 10^{-06}$ | 0.001 | $1.220 \times 10^{-06}$ | 0.890 | 0.555 |
| Rajim | 0.003 | $6.330 \times 10^{-06}$ | 0.002 | $6.550 \times 10^{-06}$ | 0.752 | $-0.356$ |

The relationship between the observed and ANN-GA-estimated SSY is shown in the Figures 5 and 6 in the form of hydrographs and scatterplots, respectively. It is clear from the Figure 5 that the SSY was overestimated in most of the gauging stations and was also underestimated at some gauging stations. The hydrograph shows that the model-estimated SSY was close to that of actual data, except at the Bamnidih, Rajim, Kurubhata and Andhiyarakore gauging stations. Similarly, it was found that the estimated and observed SSY data were closer to 45-degree line which is represented as dotted line in the scatter plots and all data points are scattered around this line at all gauging stations except Bamnidih, Andhiyarakore and Kurubhata (Figure 6). It is also observed from the scatterplots that negative sediment yield value were estimated by the ANN-GA model during low SSY data, which is unrealistic in nature. By observing the hydrographs (Figure 5a), it can be found that ANN-GA-estimated SSY data were closest to observed data at the Tikarapara station as compare to other gauging stations, with similar estimation results by the ANN model. The scatter diagram (Figure 6) of the actual and ANN-GA model-predicted values in the test data set also shows that the max points seem to be closest and lie along the 45-degree angle line, where the measured SSY values are equivalent to the estimated values. Thus, the proposed ANN-GA model was very effective for SSY at Tikarapara compared to the other gauging stations. Similarly, the proposed ANN-GA model exhibited satisfactory prediction results at the Jondhra, Sundergarh, Kantamal, Kurubhata, Baronda and Basantpur gauging stations.

It was noticed that the higher number of negative values was found by the ANN-GA model at Bamnidih, Simga and Andhiyarakore over other gauging stations (Figure 5c,e and Figure 6c,e). A similar estimation result was also observed by the ANN model. Thus, the ANN-GA model did not provide satisfactory performances between the predicted and observed SSY at Bamnidih, Simga, and Andhiyarakore. Andhiyarakore and Kurubhata are two small tributaries with relatively small catchment areas; however, they carry relatively high suspended sediment yields. This is because smaller basins are unable to accumulate sediments, making it possible to remove all eroded material [108]. The ANN-GA model was ineffective at Bamnidih. This could be attributed to the presence of the large Minimata Bango Dam, which is situated upstream of the Bamnidih station. Dams capture a huge amount of SSY [29,109]. Simga has a significantly larger CA that is dominated by limestone and seems to have a plain topographic feature. As a result, the Q and SSY were low when compared to several other tributaries with small catchment areas, such as Tel and Seonath.

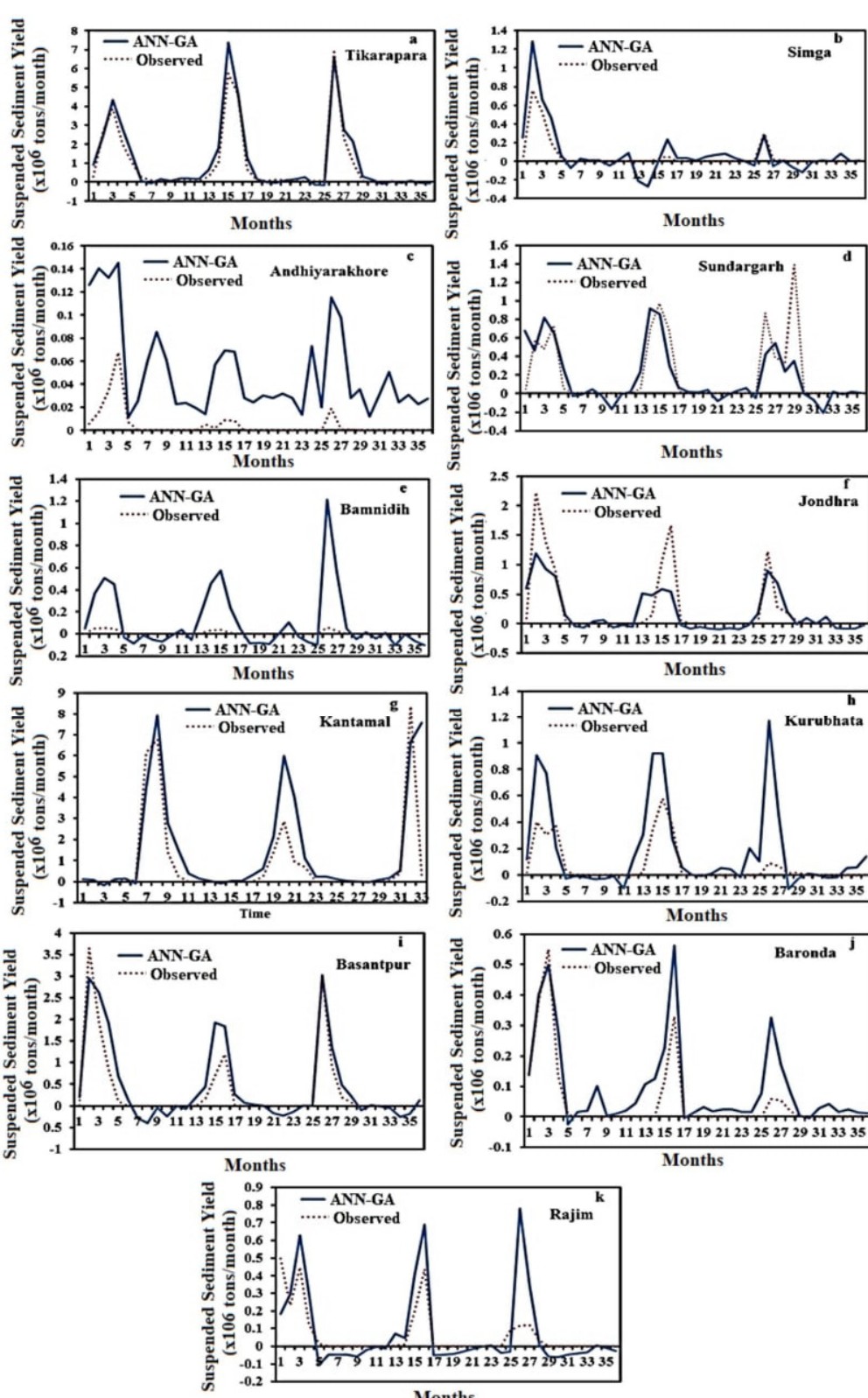

**Figure 5.** Hydrograph of the observed and ANN-GA-predicted SSY in the testing phase (**a–k**).

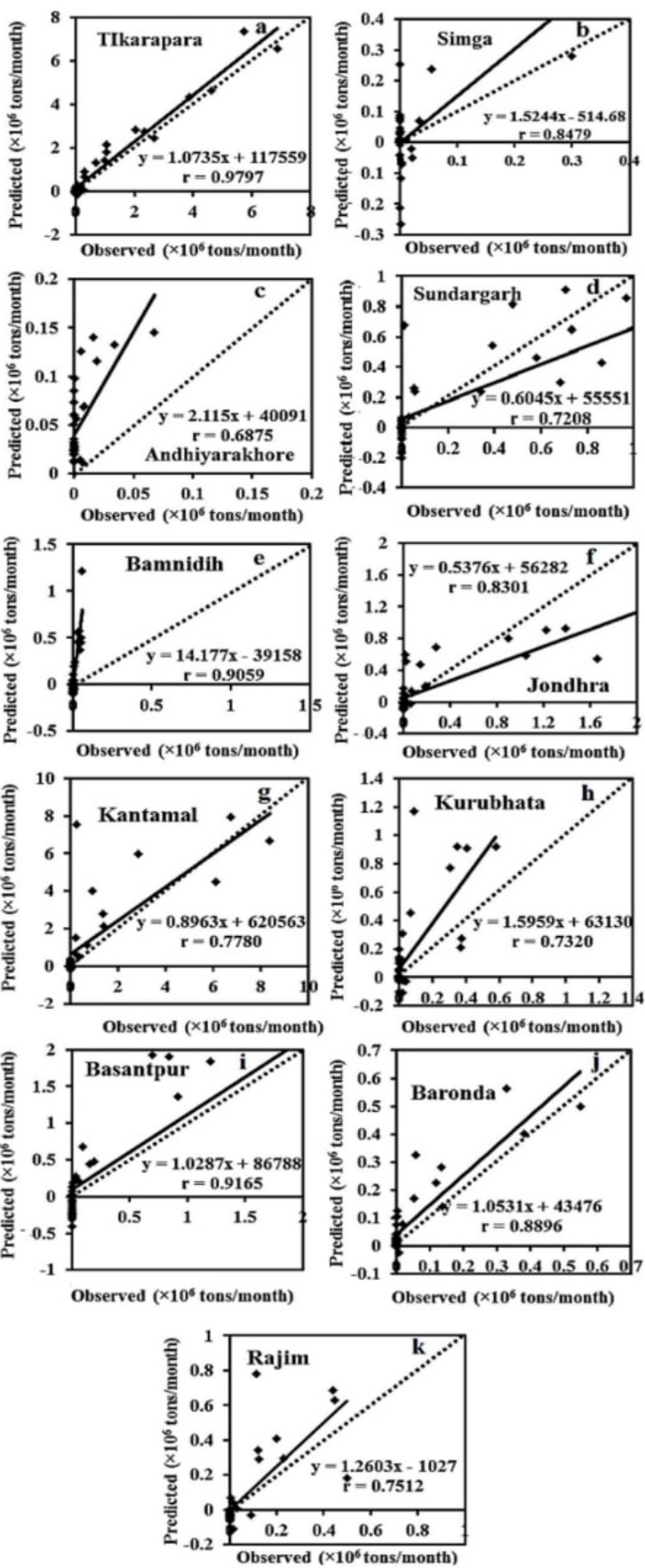

**Figure 6.** Scatterplot of the observed and ANN-GA-predicted SSY in the testing phase (**a–k**).

The observed SSY and corresponding Q, RF and T values at peaks are given in Table 4, and corresponding peaks in the hydrograph are shown in Figure 5. The Sundargarh gauge station has lowest Q and RF at time index 29 (October 2009), or at the fifth peak but the highest observed sediment load among all peaks. This disproportional change may be due to the complex nonlinear relationship and/or other factor, such as agriculture, deforesting, reforesting and road building— all examples of human activities that cause Earth's surface changes [37,110,111]—or another unknown factor. These were not considered in input list due to the unavailability of data. The proposed model was not estimated close to the observed SSY at this peak because it highly correlated to Q and other influential parameters. Similarly, Kurubhata had lowest Q and SSY at the time index 26, or the third peak (July 2009), but the highest RF. In addition, Jondhra had the lowest Q and RF with second highest sediment load at time index 16 or at the second peak (September 2008). Baronda had highest sediment load at time index 3, or at the first peak (July 2009), but the minimum Q and maximum RF. Similarly, Bamnidih contained minimum Q and maximum RF and SSY at time index 26, or at the third peak (July 2009). Time index 3 had maximum Q and minimum RF but the SSY was second highest. The model did not provide satisfactory performance at other peaks. This may have beendue to construction of the Bango Dam, which decreased the correlation between the Q and SSY. The SSY showed a good positive value of coefficient of correlation (r = 0.921) with Q for the MR at Tikarapara and other gauging stations, except Bamnidhi (r = 0.591) [31]. The time index 4 (September 2007) of Andhiyarakhore provided the highest SSY and Q with second lowest RF. The lowest SSY was found at the time index 4 (July 2009) with the corresponding highest Q and second highest RF. The complex nonlinearity occurred due to smallest catchment area. At other peaks, the estimated and observed SSY were not disclosed. The ANN-GA model was not able to estimate the SSY closest to the observed SSY at some peaks due to very complex nonlinear temporal and spatial hydro-climatically and geo-morphological conditions. The ANN-GA model provided best result at the Tikarapara station.

**Table 4.** Water discharge, rainfall and temperature hydro-climatic data corresponding to peaks of suspended sediment yieldin Mahanadi River Basin.

| Gauge Station | Time Index of Peaks | Q (m³/s) | RF (mm) | Sediment Yield (Tons/Month) | T (°C) |
|---|---|---|---|---|---|
| | 2 | 7756 | 351 | 582,190 | 27 |
| | 4 | 11,857 | 242 | 732,762 | 29.5 |
| Sundargarh | 15 | 11,239 | 488 | 970,420 | 27.5 |
| | 26 | 8164 | 591 | 863,698 | 28.5 |
| | **29** | 6488 | 169 | 1,386,340 | 26 |
| | 2 | 4551 | 406 | 403,901 | 29.5 |
| Kurubhata | 4 | 6463 | 207 | 371,259 | 31 |
| | 15 | 9845 | 529 | 578,737 | 29 |
| | **26** | 3130 | 566 | 90,388 | 26.5 |
| | 2 | 42,770 | 404 | 2,225,625 | 30.5 |
| Jondhra | **16** | 19,207 | 152 | 1,660,837 | 30.75 |
| | 26 | 25,906 | 508 | 1,226,185 | 30.75 |
| | 3 | 12,363 | 245 | 550,873 | 29.25 |
| Baronda | 15 | 13,228 | 267 | 330,656 | 28.75 |
| | **26** | 2668 | 592 | 57,911 | 29.25 |
| | 3 | 6138 | 428 | 53,056 | 29.5 |
| Baminidhi | 15 | 5877 | 441 | 40,044 | 29 |
| | **26** | 3256 | 636 | 58,940 | 34 |
| | **4** | 1244 | 233 | 67,731 | 28.5 |
| Andhiyarakhore | 15 | 325 | 139 | 8615 | 28 |
| | 26 | 220 | 334 | 19,228 | 31.5 |

### 4.3. Comparisons among ANN-GA, SRC, MLR and ANN Models

After developing a robust model, the model's evaluation was performed using testing data that were not used in the training stage. The prediction capability of artificial intelligence models was evaluated by comparing their results to classic regression approaches.

For a comparison analysis, all models used the same test data set. The comparison was performed using the test data's estimated values. Figure 7 shows the error statistics (r and RMSE values) of the SRC, ANN-GA, MLR and ANN models during the testing phase. The RMSE (0.00875) of the ANN-GA model was lower than the RMSE (0.00892) of the ANN model in the testing stage due to the single objective GA attempted apply to the ANN model for optimizing all ANN parameters concurrently. When compared to the traditional ANN model, the ANN-GA models lowered the error by 1.85 percent. By taking into account the optimal input variables and associated parameters of the ANN, the ANN-GA model outperformed the ANN model. The error was reduced when the GA was used for optimizing the all parameters of the ANN simulataneously. The concurrently optimization of all ANN variables using the single objective GA revealed this superiority. When compared with traditional SRC, the ANN and ANN-GA models diminished error by 11.89 percent and 15.35 percent, respectively. Similarly, when compared to the traditional MLR model, the ANN-GA and ANN models reduced error by 2.334 percent and 0.491 percent, respectively. All intelligence-based models (ANN-GA and ANN) outperformed conventional approaches using RMSE and r as performance criteria. It was revealed that ANN-GA model had lowest RMSE and highest r among all comparative models (Figure 7). However, the SRC model had the highest RMSE and lowest r among all models. As a result, the SRC model had the lowest predictability, while the ANN-GA model had the best predictability. It was observed that the the ANN-GA model outperformed the SRC, MLR and ANN models. Recently, it was also observed in hydrological studies and other fields that the hybrid model, i.e., GA-based ANN, provided better prediction results compared to traditional ANN, MLR and SRC models [61,69,70]. The results suggested that in comparison to other classic MLR and SRC models, the ANN model outperformed them. Similarly, research has shown that the ANN technique outperformed the conventional SRC and MLR methods [17,25,39,112,113].

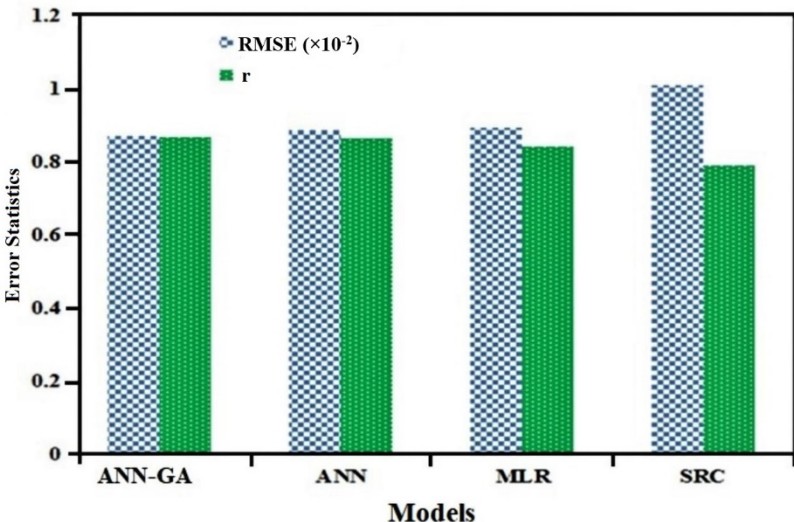

**Figure 7.** Variation of RMSE and r of ANN-GA, ANN, MLR and SRC models.

Thes findings demonstrate that the ANN-GA model performed well and had better generalization potential. This superiority was achieved through the optimization of all ANN parameters simultaneously using GA. In addition, when it comes to estimating SSY in the MR basin, the ANN and ANN-GA models outperformed the MLR and SRC models. The optimization of the ANN parameters using the GA is a superior strategy than the usual trial-and-error and grid-searching methods for the model parameters selection. This

research indicates that the chosen parameter using these approaches not only enhances the model's performance but also greatly decreases computational time.

Figure 8 depicts the error histograms of all four models used to assess the uncertainty of all proposed models. These histograms show the dataset's unexplained variations, which cannot be describe by the models. As a result, unexplained variability in the model can be regarded as uncertainty. Uncertainty can be defined as knowledge situations involving incomplete or unknown information [114]. To quantify uncertainty, three sources should be considered: physical variability of equipment, data variability and model error [115,116]. Method uncertainty analysis is described in the context of various factors, such as input variability, measurement errors, assumptions and approximations and sparse and imprecise data during the modelling process [117]. In statistical techniques, uncertainty analysis is typically based on the estimated variance and confidence limits by assuming a normally distributed error, which has been well described in the literature [118–122]. All histograms are distributed normally, with a mean close to zero. These findings demonstrate that the uncertainties were not biased (approximately zero mean) and had a Gaussian distribution. The uncertainty quantification from Gaussian distributions can be achieved through the variance of the error. The ANN-GA model had the least uncertainty, although the SRC model had the most (greatest and least variance are found in SRC and ANN-GA models, respectively). These findings also suggest that some ambiguities in the SSY data remain that cannot be modelled by utilizing the considered factors in this study. Human activities, runoff, damming factor, geomorphology and other factors could have influenced the SSY model. Our future research will take into account the effects of these factors.

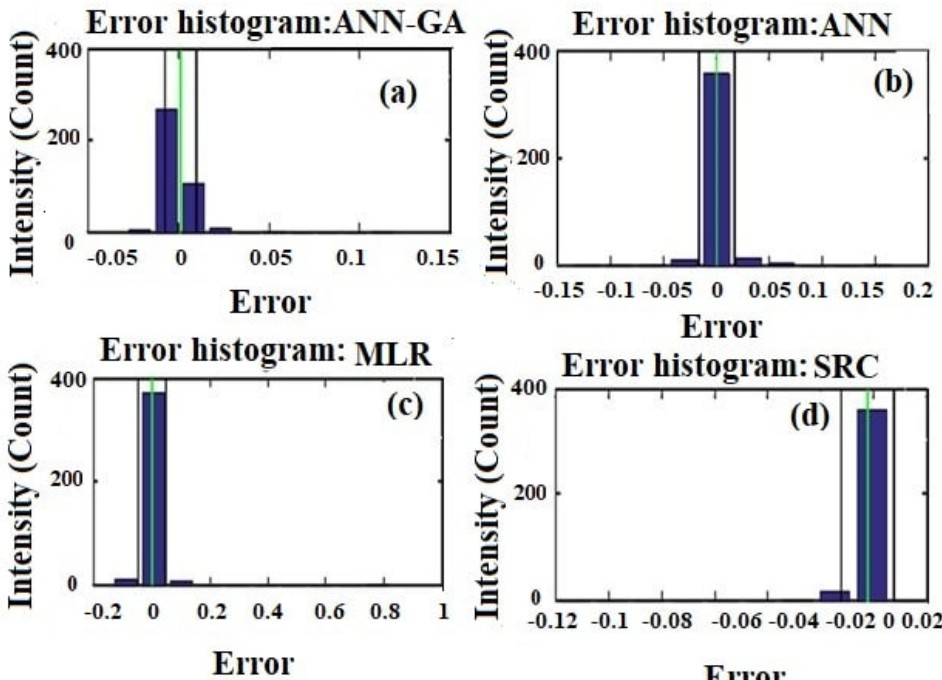

**Figure 8.** Error histogram of models: (**a**) ANN-GA, (**b**) ANN, (**c**) MLR, (**d**) SRC.

## 5. Conclusions

The ANN-GA model was used to estimate SSY in the Mahanadi River. This study was based on temporal data, such as water discharge, temperature, rainfall, and suspended sediment yield and spatial data rock type, relief and catchment area. These data were used as input variables in the SSY estimation model using data from 11 gauging stations. The results showed that Q and SSY had a wide spatial variability in Mahanadi River Basin. Tikarapara had the highest water discharge, rainfall, catchment area and supended sediment yield values, while Andhiyakore had the lowest. Considering the above factors, the

hybrid ANN-GA estimation model efficiently optimized multiple ANN model parameters concurrently using GA for SSY prediction.

The scatterplot between observed and modelled estimated SSY using ANN-GA indicates that ANN-GA outperformed the ANN, SRC and MLR estimation models. The ANN-GA is the best model for SSY prediction in the Mahanadi River based on RMSE, MSE, MAE, correlation coefficient and coefficient of efficiency. Furthermore, both artificial intelligence models (ANN and ANN-GA) outperformed the SRC and MLR models for SSY prediction. It is also concluded that the ANN-GA calculated SSY successfully in the sub-basin with the largest catchment area and delivered the best results at the Tikarapara gauge station, which has the largest catchment area. Model performance was inferior in stations with small catchment areas, with the lowest model performance observed at Andhiyarakhore, which has the smallest catchment area in the Mahanadi River. Furthermore, most models did not accurately estimate SSY at Bamnidih, Andhiyarakore, Kurubhata and Rajim. The developed single ANN-GA model has more substantial generalisation potential to predict SSY at Tikarapara due to the inclusion of training utilizing data from all gauge stations of the Mahanadi River Basin instead of a single gauge location and concurrently optimisation of all ANN parameters. This method will be necessary for better water management in the MR, India, as well as the geomorphology, construction of dams, canals, bridges, piping, streams, water treatment methods and the assessment of water quality issues. This analysis excludes data on rainfall intensity, land use and land cover changes and other anthropogenic factors. As a result, future studies will include these factors.

**Author Contributions:** Conceptualization, A.Y., M.K.H. and D.J.; Data curation, V.K. and H.M.; Formal analysis, A.Y., M.K.H., D.J., V.K. and M.S.A.; Funding acquisition, M.K.H.; Investigation, A.Y., D.J. and H.M.; Methodology, A.Y., D.J. and V.K.; Resources, A.Y., D.J., H.M. and H.A.; Software, A.Y. and D.J.; Validation, M.K.H.; Writing—original draft, A.Y.; Writing—review & editing, M.K.H., A.H.M.A., V.K., H.M., H.A. and M.S.A. All authors have read and agreed to the published version of the manuscript.

**Funding:** This paper was supported by the University Kebangsaan Malaysia Under FRGS/1/2020/ICT03/UKM/02/6.

**Institutional Review Board Statement:** Not applicable.

**Informed Consent Statement:** Not applicable.

**Data Availability Statement:** The data used in the study were obtained after signing a non-disclosure agreement with the Central Water Commission (CWC). As a result, these data cannot be shared. However, the solutions can be shared. Please contact the first author with proper justification.

**Acknowledgments:** This research was supported by the Universiti Kebangsaan Malaysia under research grant FRGS/1/2020/ICT03/UKM/02/6. Also, this research was was supported by Taif University Researchers Supporting Project number (TURSP-2020/216), Taif University, Taif, Saudi Arabia. The others data support by the Government organization (India-WRIS) of India. The author appreciates to the National Institute of Technology, Rourklea and Koneru Lakshmaiah Educational Foundation, India, for offering the necessary facilities for this research.

**Conflicts of Interest:** The authors declare no conflict of interest.

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
