# Peer review of "Optimized Scenario for Estimating Suspended Sediment Yield Using an Artificial Neural Network Coupled with a Genetic Algorithm"

_water, doi:10.3390/w14182815_

Round 1

Reviewer 1 Report (New Reviewer)

The manuscript prented a hot topic of machine learning applicaiton in hydrological science. In genenral, more discussions are needed, some figures presented very simple info, which need improvement.

detailed comments please see the attachment

Author Response

Reviewer 1-Comments-answer

Comment: The manuscript parented a hot topic of machine learning application in hydrological science. In general, more discussions are needed, some figures presented very simple info, which need improvement.

Response: Thank you for your valuable comments. We have revised the manuscript as per your valuable comments.

Comment 1: Introduction is too long. what is the research gap, or what is the novelty of the work?

Response: Thank you for your valuable comments. As per suggestions of the other reviewers, many lines are already added in the introduction section so introduction is long. Our main goal is to demonstrate the fact that simultaneous selection of all parameters including the input variable can significantly improve the robustness of the model and performance of the model. In this study, we have considered and presented a artificial neural network model where all parameters (inputs variables, hidden layer neurons, combination coefficient, transfer function, and network weights) are optimized simultaneously, and applied for sediment yield estimation from a river. Several research shows that this approach is not only reduce the computational intensiveness, but also provide superior results as compare to trial error approaches and grid search approaches. Few researchers applied ANN and GA algorithm for predicting sediment at specific single gauging stations in Mahanadi river. The study of Mahanadi River provided some insight into the application of artificial intelligence method in sediment estimation in the Mahanadi river system by using a huge amount of data of eleven gauging stations in a single generalized model and applied it on individual gauging stations for estimation of sediment yield. To the best of the knowledge of authors, there is no such study available for sediment yield estimation in the Mahanadi river basin. The key innovation of this research is that it developed a fully automatic highly generalized global hybrid Artificial Intelligence based model i.e. ANN-GA for prediction of sediment yield in Mahanadi river, where minimum human intervention is required.

      Artificial intelligence and machine learning algorithm are, now a day, widely using in many fields. These techniques have significant number of advantages to deal with complex data, and that has been well documented in the different literature. However, many of these techniques are either non-linear in nature or need to map in high dimensional space. Due to this reason, these methods are always associated with large number of parameters which need to be selected carefully for desire performance of the model. Generally, the parameters for these methods are selected by trial-and-error and/or grid search method to get a reasonably good result. However, this approach takes significantly large amount of computational time to come up with the parameters value, and not also guaranteed optimal or near-optimal solution of the problems. Different applications of artificial intelligence and machine learning models demonstrated the use of heuristic and meta-heuristic approach for simultaneous optimization of associated parameters in artificial intelligence models to overcome the limitation of trial-and-error methods. Several research shows that this approach is not only reduce the computational intensiveness, but also provide superior results.

      According to authors, in the field of hydrology for sediment yield prediction, this simultaneous optimization of parameters have never been tested and applied. In sediment yield prediction, heuristic and meta-heuristic are combined with artificial intelligence models; however, not all parameters are optimized simultaneously in those study (Rajaee et al. 2009; Sahoo and Jha 2013; Kisi 2005; Zhu et al. 2007, 2008, Yadav et. al. 2017). We are adopting the technique of using genetic algorithm as a meta-heuristic algorithm to optimize all parameters in artificial intelligence models (artificial neural network) for sediment yield prediction in the Mahanadi river basin system. Our methodology may not be unique, but its application in sediment load prediction is unique. Moreover, the case study of Mahanadi River provides some insight into the application of artificial intelligence and machine learning methods in sediment prediction in river system. We have revised our manuscript and highlighted our main contribution. We have made correction and updated it in the revised manuscript.

References

Sahoo S, Jha MK. Groundwater-level prediction using multiple linear regression and artificial neural network techniques: a comparative assessment. Hydrogeology Journal. 2013 Dec;21(8):1865-87.

Kisi, O. Suspended sediment estimation using neuro-fuzzy and neural network approaches. Hydrol Sci J 2005, 50(4): 683–696.

Zhu, Y.M.; Lu, X.X.; Zhou, Y., Suspended sediment flux modeling with artificial neural network: an example of the Long-chuanjiang River in the Upper Yangtze Catchment, China. Geomorphology, 2007, 84(1–2), 111–125.

Zhu, Y.-M.; Lu, X. X.; Zhou, Y., Sediment flux sensitivity to climate change: a case study in the Longchuanjiang catchment of the upper Yangtze River, China. Global Planetary Change, 2008, 60, 429–442.

Yadav, A.; Chatterjee, S.; Equeenuddin, S.M., Prediction of Suspended Sediment Yield by Artificial Neural Network and Traditional Mathematical Model in Mahanadi River Basin, India, Journal of Sustainable Water Resource Management, 2017, 4(4), 745-759.

Comment 2: The study of SSY estimation is based on MR. Revise this sentence.

Response: Thank you for your valuable comments. We have corrected this sentence in the revised manuscript. The following lines are updated in the revised manuscript.

Mahanadi river is selected for the study of suspended sediment yield estimation.

Comment 2: Tikarapara station has highest value of catchment area (124450 km2) which is present in downstream whereas Andhiyarakore has lowest catchment area (2210 km2) which is situated in upstream of the MR basin. check the spelling errors and grammar.

Response: Thank you for your valuable comments. We have corrected this sentence in the revised manuscript. The following lines are updated in the revised manuscript.

The maximum drainage area of 124450 km2 is covered by Tikarapara (farthest downstream station) while Andhiyarakhore (upstream station) covers minimum drainage area of 1100 km2 in the MR basin.

Comment 3: Line 178-183- During the study it has found that the average annual RF varies from 1200-1400 mm 178 in the year 1971-2004 [74]. The annual RF of 90% is found at the third quarter of the month which being monsoon period for Mahanadi basin. The fourth quarter of the year is coldest, nearly 12°C minimum T. The highest T rises from 39°C to 40°C in second quarter of the year recorded between 1969 and 2004[74]. Humidity ranges from 68 to 87% high and 9 to 182 45% low. In average humidity of the basin is 82% high and 31.6% low [80]. use rainfall not RF. revise ling 178-183.

Response: Thank you for your valuable comments. We have corrected the sentences in the revised manuscript. The following lines are updated in the revised manuscript.

The average annual RF varies in entire MR basin from 1200-1400 mm for the period 1971-2004 [74]. The MR basin receives nearly 90% of the annual rainfall during monsoon period (June to October). The coldest months of the year are December and January having lowest temperature of 12°C whereas April and May registers the highest temperature ranging from 39°C to 40°C for the period 1969-2004 on the basis of daily data [74]. The maximum and lowest relative humidity range is 68 to 87% and 9 to 45%, respectively. The average highest and minimum relative humidity of the basin is 82% and 31.6% respectively [80].

Comment 4: At some sites, the monthly data were not well simulated, explain why. Some peaks were overestimated, for example in Figure 7e and 7h, 7k.

Response: Thank you for your valuable comments. It is observed from the Figure 4 and Figure 5 that higher number of negative value is generated by model at low sediment value and overestimated at Bamnidih and Andhiyarakhore as comparative other gauging stations; however, the suspended sediment yield can never be negative in reality. One author of this paper also got negative sediment value where sediment is low or close to zero which is described by Yadav et al. (2017, 2018, 2021) and Yadav and Satyanarayana(2020). Similar negative value of the suspended sediment predicted value using artificial intelligence models was also obtained in many river basins (Zhou et al. 2007; Yadav et al. 2017, 2018; Yadav and Satyanarayana 2020). This demonstrated the facts that the data has strong non-linear behaviour in the area of small, valued samples of suspended sediment yield. Addition to it, complex non-linear erosion and transportation process of sedimentation which caused poor performance of the proposed ANN model (Zhu et al. 2007; Rajaee et al. 2009; Malesse 2011; Kisi and Shree 2012). One author of this paper also got negative sediment value where sediment is low or close to zero which is described by Yadav et al. (2017, 2018, 2021) and Yadav and Satyanarayana(2020). Similar negative value of the suspended sediment predicted value using artificial intelligence models was also obtained in many river basins (Zhou et al. 2007; Yadav et al. 2017, 2018; Yadav and Satyanarayana 2020). This demonstrated the facts that the data has strong non-linear behaviour in the area of small, valued samples of suspended sediment yield. Addition to it, complex non-linear erosion and transportation process of sedimentation which caused poor performance of the proposed ANN model (Zhu et al. 2007; Rajaee et al. 2009; Malesse 2011; Kisi and Shree 2012).This demonstrates the fact that the data has strong non-linear behaviour in the area of small valued samples. The proposed model fails to capture that complex nonlinearity which results in some negative estimated values.

         It is also observed from scatter plot that very large deviation occurs between the regression line and bisector line at Bamnidih. It may be due to the presence of the large Minimata Bango dam. It is seen from the scatter plots and hydrographs that proposed model is not also providing satisfactorily result at Andhiyarakore. It may be due to the lowest catchment area. Addition to it, complex non-linear erosion and transportation process of sedimentation which caused poor performance of the proposed GA-ANN model (Zhu et al. 2007; Rajaee et al. 2009; Malesse 2011; Kisi and Shree 2012). In both Andhiyarakore and Bamnidih station, the modelled suspended sediment yield show large variation in all peaks and corresponds to low value of observed suspended sediment yield as compare to other gauging stations.

        Poor performance of the GA-ANN model in predicting suspended sediment at Bamnidih, Andhiyarakore, Kurubhata and Rajim station can be attributed to the complex interaction of various controlling factors of sediment yield. Andhiyarakore, Kurubhata and are two small tributaries having relatively small catchment area; however they carry relatively high suspended sediment. It is due to the inability of smaller basins to store sediments and allow all the material that is eroded to be removed (Chakrapani and Subramanian, 1990). As mentioned earlier, at Bamnidih the presence of dam is the major factor for deviation of the modelled output. Though Simga has relatively large catchment area, it has very flat topography and dominated by limestone. Thus, the water discharge and sediment yield is low in comparison to other tributaries – Tel and Seonath that have small catchment area.

Comment 5: Figure 8i-when we looked at these figures, it can be found that the simulation results were not good enough. explain why?

Response: Thank you for your comments. The proposed model is provided good results for moderate and high sediment value, but it is not provided good performance for the very low sediment value. The proposed model has generated many numbers of negative sediment yield value at very low sediment value or close to zero sediment value but sediment can not be negative which is unrealistic. This demonstrates the fact that the data has strong non-linear behaviour in the area of small valued samples. It indicates that highly complex nonlinear sedimentations process has occurred at small valued sample location of sediment yield which is not being captured by the model and generated negative suspended sediment yield values. The bisector line and regression line are close to each other for moderate and high sediment value.

The complex non-linear erosion and transportation process of sedimentation which caused poor performance of the proposed GA-ANN model at this station (Zhu et al. 2007; Rajaee et al. 2009; Malesse 2011; Kisi and Shree 2012).

Comment 6: Make sure the RMSE and r esimated correctly.

Response: Thank you for your comments. It is correct.

Comment 7: More discussions are needed.

Response: Thank you for your comments. The following lines are added in the revised manuscript.

Uncertainty can be defined as knowledge situations involving incomplete or unknown information (Ga rdenfors & Sahlin, 1982). To quantify uncertainty, three sources should be considered: physical variability of equipment, data variability, and model error (Barford, 1985; Kennedy & O'Hagan, 2001). Methods uncertainty analysis are described in the context of various factors such as input variability, measurement errors, assumptions and approximations, and sparse and imprecise data during the modelling process (Yan et al., 2015). In statistical techniques, uncertainty analysis is typically based on the estimated variance and confidence limits by assuming normally distributed error which are well described in various literatures (Cacuci & Ionescu-Bujor, 2004; Jiang et al., 2018; Zhang et al., 2020).

Reviewer 2 Report (New Reviewer)

Dear authors,

I have enjoyed reading and reviewing your manuscript entitled: "Optimized scenario for estimating suspended sediment yield using an artificial neural network coupled with Genetic algorithm".

The topic is very interesting and the approach followed by the authors is promising, providing with robust results. As such, I think that the manuscript is worth publication in the Journal Water.

Nevertheless, I think that the manuscript needs substantial improvements. I have provided the authors in the attached .pdf with quite some comments that may help for that task. I summarize here below the main points, which in my view merit some more elaboration:

(1) In the introduction, I would detail more the variables that are the most important to predict the suspended sediment yield, based on appropriate literature, and explain how they affect it. It should be shown that the variables picked in the model are the most appropriate, and not justify their choice simply by a matter of data availability. Secondary variables not taken into account in the model should also be mentionned, and their potential effect on the model accuracy assessed in the Discussion.

(2) It is explained nowhere how the different variables are measured. I am aware that there are 11 stations, with potential differing measuring systems, but I think it is important to have an idea on how these variables are measured, sine their measurement is not deprived of uncertainty neither. I would add a small paragraph in the methods that summarizes how the different variables are measured, and the detailled monitoring systems could be provided in a Supporting Information, or referenced to the appropriate literature if existing. The uncertainty around the variable measurement and its effect on the model results could be further assessed in the Discussion.

(3) It was not always clear to me how the different training, validation and testing subsets were used, and why one or the other was presented in the results. The purpose of the different data subsets and use could be made clearer in the Methods and Results sections.

(4) I would considerably change the way the main results are presented. Figures 2, 3, 4 and 5 could be grouped into a single map showing how the different variables vary spatially across the watershed. Perhaps histograms showing values of RF, Q, T and SSY for each measuring station on a map would be a nice way to visualize how these variables vary across the watershed? Doing so, Table 2, which basically provides with the same numbers, could go in a Supporting Information. I think it may also worth adding another map Figure, which would delineate the different subcatchment area and show the spatial distribution of rock types, since these are also two variables included into the model.  I am also wondering whether Table 3 is necessary. Little of its content is described at this stage in the text, and perhaps discussing the average correlation between SSY and Q-T-RF measured over the 11 measuring station would be enough, without the need for a Table with the detailed correlation for each site. If the authors think that the Table is necessary, it should be described into more details in the text.

(5) From the results, the ANN-GA does globally provide with lower error and higher correlation, as compared to the other more traditional methods (e.g. SCR, MLR), but sometimes I have the feeling that the results may be a little overrated. For instance, the descrease in RMSE and increase in correlation between ANN-GA and MLR is not that large from what I see. It would be good to add some room in the discussion, either to show with tangible numbers that the improvement is substantial, or assess whether the gain of ANN-GA approach worthes the effort with regards to a simpler MLR approach.

(6) There is quite some improvements needed in the general writing and phrasing. Some sentences were hard to follow, and I think the manuscript would benefit from a general revision of the text. I have pointed out in the attached .pdf where I found the text could be improved. There is also a subtantial use of acronyms throughout the manuscript, which turn sometimes the reading difficult. As such, I would remove substantially the amount of acronyms used (e.g. variable names, Mahanadi river).

Hope my review will help the authors to improve this very interesting piece of work.

#End of review.

Author Response

Reviewer 2-Comments-answer

General comments

Comments: Nevertheless, I think that the manuscript needs substantial improvements. I have provided the authors in the attached .pdf with quite some comments that may help for that task.

Response: Thank you for your valuable comments. We have corrected it. As per your comments and suggestions, we have updated it in the revised manuscript.

Comment 1: In the introduction, I would detail more the variables that are the most important to predict the suspended sediment yield, based on appropriate literature, and explain how they affect it. It should be shown that the variables picked in the model are the most appropriate, and not justify their choice simply by a matter of data availability. Secondary variables not taken into account in the model should also be mentioned, and their potential effect on the model accuracy assessed in the discussion.

Response: Thank you for your valuable comments. Climate change is expected to impact on hydrological processes and water resources (Mileham et al., 2009; Labat, 2010). The effect of rainfall splash detachment and entrainment through overland flow generates sediments. Coulthard et al. (2000) indicated that the sediment flux increases with the increase in rainfall as well as runoff. Zhu et al. (2007) and Ghose et al. (2012) also reported that temperature plays a secondary role in sediment yield or erosion rate, which are the dominant driving force of sediment generation and sediment transportation. Temperature controls the sediment yield in several indirect ways. The change in temperature may affect the sediment discharge by altering runoff and changing the erosion rate through its influence on evapotranspiration, vegetation and weathering (Zhu et al. 2007, 2008).

The majority of water discharge in the river is contributed from the precipitation and groundwater recharge accounts for small contribution. Good non-linear relationship between sediment yield and water discharge has been reported in most of the world’s major river (Wood 1977; Ramesh and Subramanian 1988; Vaithiyanathan et al. 1988; Mossa 1990; Gupta and Chakarpani 2005; Yadav et al. 2017; Bastia and Equeenuddin 2017). The influence of river basin characteristics such as geology, soil and relief on sediment yield was investigated by Jansson (1982).  Basin relief is the most dominating factor for mechanical denudation in basin which causes high rate of erosion and sediment load. Slope gradient is one of the major factors affecting soil particle detachment and transport (Fu et al. 2011). Basin area influences sediment yield due to variation in catchment properties like gradient, storage capacity etc. (Chakrapani 2005). Rock type is important controlling factor for the erosion. The sediment load in river is produced by physical and chemical weathering of the rock and soil within basin.

     Mining activities are responsible for supply of sediment to nearby streams. Vegetation above ground surface decreases the runoff and erosion by weakening the gravitational energy of raindrops and vegetation canopy-cover protects the soil from direct impact of raindrops (Dou 1975). Wilson (1972) demonstrated that land use as an important controlling factor of sediment yield. Human activities related to land surface disturbance such as deforestation, afforestation, agriculture activity, urbanization, mining activites and road construction played an important role in enhancing the suspended sediment (Nelson and Booth 2002; Zhou et al. 2004; Lu 2005; Chakrapani 2005; Restrepo and Syvitski 2006). The secondary variable which are not used in this study. These variable will be considered for future research. The following lines are added in the revised manuscript.

The majority of the water discharge in the river is caused by precipitation. Sedimentation is caused by the effects of rainfall splash detachment and entrainment through overland flow. Temperature changes may influence sediment discharge by altering runoff and changing the rate of erosion due to their effects on evapotranspiration, vegetation, and weathering (Zhu et al. 2007, 2008). The most dominant factor for mechanical denudation in a basin is basin relief, which causes a high rate of erosion and sediment load. One of the major factors influencing soil particle detachment and transport is slope gradient (Fu et al. 2011). The variation in catchment properties such as gradient, storage capacity, and so on influences sediment yield (Chakrapani 2005). The type of rock is an important controlling factor for erosion.

Comment 2: It is explained nowhere how the different variables are measured. I am aware that there are 11 stations, with potential differing measuring systems, but I think it is important to have an idea on how these variables are measured, sine their measurement is not deprived of uncertainty neither. I would add a small paragraph in the methods that summarizes how the different variables are measured, and the detailled monitoring systems could be provided in a Supporting Information, or referenced to the appropriate literature if existing. The uncertainty around the variable measurement and its effect on the model results could be further assessed in the Discussion.

Response: Thank you for your valuable comments. We have collected the all data form Central Water Commission (CWC), Bhubaneswar, Odisha. The data measurement is done by CWC team members. The details descriptions about the data measures are given many literatures (CWC 2012, India-WRIS 2015). We have corrected the manuscript. The following lines are added in the revised manuscript.  

Uncertainty can simply be described as knowledge situations involving imperfect or unknown information (Ga¨rdenfors & Sahlin, 1982). To quantify uncertainty, three sources including physical variability of equipment, data, and model error should be considered (Barford, 1985; Kennedy & O’Hagan, 2001). In the modeling process, methods UQ are described in the context of different factors, such as input variabilities, assumptions and approximations, measurement errors, and sparse and imprecise data (Yan et al., 2015).

References

CWC (Central Water Commission). (2012). Integrated hydrological data book. Hydrological data directorate, information systems organization, water planning and projects wing. In Central water commission. New Delhi: Hydrological Data Directorate, Information System Organization, Water Planning and Projects Wing, Central Water Commission (CWC).

India-WRIS. (2015). Water resources information system of India. http://india-wris. nrsc.gov.in/wrpinfo/index.php?title¼Mahanadi. (Accessed 5 May 2017).

Comment 3: It was not always clear to me how the different training, validation and testing subsets were used, and why one or the other was presented in the results. The purpose of the different data subsets and use could be made clearer in the Methods and Results sections.

Response: Thank you for your valuable comments and suggestions. Thank you for your valuable comments. Data are divided into training (70 percent), validation (15 percent) and testing (15 per-cent) [25,36,58,70,84]. The training data are taken from June 01, 1990, to May 31, 2004, and validation data are taken from June 1, 2004, to May 31, 2007 and similarly for testing purpose data are listed from June 01, 2007 to May 31, 2010. We have given same training, validation and testing data for all models. We have taken starting 70% data for training, next 15% for validation and next 15 % for testing. If data is not fixed for training, validation and testing, then it is difficult to compare the performance of the models using different testing data set. Here, we have given same testing data to all models and compare the performances. If we will not fixed the data then model will select different data set as per division of data and testing data may be different for other comparative models but division percentage will be same. The t-test was to check the division of data is correct or not. We have corrected and updated in the revised manuscript. The following lines are added in the revised manuscript.

Data are divided into training, validation and testing. Training data (70 percent of the dataset) is used to construct the models; testing data (15 percent) is utilised in the proposed model to assess the model's performance and validation data (15 percent) is used to avoid overfitting of the developed models. Testing data is unseen data that is not used in the model development process.

Comment 4: I would considerably change the way the main results are presented. Figures 2, 3, 4 and 5 could be grouped into a single map showing how the different variables vary spatially across the watershed. Perhaps histograms showing values of RF, Q, T and SSY for each measuring station on a map would be a nice way to visualize how these variables vary across the watershed? Doing so, Table 2, which basically provides with the same numbers, could go in a Supporting Information. I think it may also worth adding another map Figure, which would delineate the different sub catchment area and show the spatial distribution of rock types, since these are also two variables included into the model.  I am also wondering whether Table 3 is necessary. Little of its content is described at this stage in the text, and perhaps discussing the average correlation between SSY and Q-T-RF measured over the 11 measuring station would be enough, without the need for a Table with the detailed correlation for each site. If the authors think that the Table is necessary, it should be described into more details in the text.

Response: Thank you for your valuable comments and suggestions. All data sets have wide range. These data have different dimensions and different units. If plot in single map, then all data points can not be visible clearly due to wide range of different data sets. We combined all Figures 2, 3,4 and 5 in one figure. We have described more details about the Table 3 in the revised manuscript. This table is suggested by one reviewer. So reason, we have added this Table 3.In future research, we will consider the your other valuable suggestions.

Comment 5: From the results, the ANN-GA does globally provide with lower error and higher correlation, as compared to the other more traditional methods (e.g. SCR, MLR), but sometimes I have the feeling that the results may be a little overrated. For instance, the descrease in RMSE and increase in correlation between ANN-GA and MLR is not that large from what I see. It would be good to add some room in the discussion, either to show with tangible numbers that the improvement is substantial, or assess whether the gain of ANN-GA approach worthes the effort with regards to a simpler MLR approach.

Response: Thank you for your valuable comments and suggestions. Authors do agree with the comment raised by the reviewer that artificial intelligent based methods like ANN-GA provide better result than the traditional methods. However, it is not always the case. Kisi (2012) developed ANN based discharge-suspended sediment relationship models for the Eel river, California, and observed that the traditional sediment rating curve (SRC) model provided better performances than the artificial intelligent based methods in the downstream.  The MLR method is linear regression approach which can not handle the complex nonlinear behaviour of the sedimentation. Sometime MLR method may also provide better result than other artificial intelligence on the basis of some conditions.

By keeping above finding in mind, We have compared our methods (ANN and ANN-GA) with multiple linear regression (MLR), and sediment rating curve (SRC) models. There are large of number of traditional methods available in the literature; however, MLR and SRC techniques are widely used traditional mathematical models in the sediment yield modelling community. Therefore, we have decided to use these two traditional mathematical models for our research. We have compared ANN, GA-ANN models with MLR and SRC models. n the Figure 9, the RMSE value of ANN-GA model is lowest among all comparative models (ANN, MLR and SRC). The differences between the GA-ANN and ANN are slightly. It is also clear from the Figure 6 that there is large difference between the performance of ANN-GA and SRC. The SRC is worst model among all models. The GA-ANN is the comparatively best model among all models. The ANN-GA model is moderately better than the other models. The key innovation of this research is that it developed a fully automatic highly generalized global hybrid Artificial Intelligence based model i.e. ANN-GA for prediction of sediment yield, where minimum human intervention is required. In the ANN-GA model, inputs parameters are selected using the GA algorithm automatically and judicially. These approach is not only reduce the computational intensiveness, but also provide superior results as compare to trial error approaches and grid search approaches.

     There is strong debate about which methods perform better. There are plenty of literature out there including number of research work of one of the co-authors of this manuscript that demonstrated the fact that both support vector machine (SVM) and ANN performs equally well, if the models are properly trained. Their performance differences are very negligible. One instance a specific method may show superiority over other methods; however, that can be changed if a different data set or application is chosen. The goal of this paper is not to show the superiority of one method (ANN) over other methods, rather we are trying to demonstrate the fact that we really need to select different modelling parameters judicially to generate a robust model. The ANN is selected just as a non-linear technique and GA is used for selecting optimum parameters of the ANN. In the MLR model, the input parameters may be select using trail error approach or greed search algorithm which is time consuming process. In the GA-ANN, all ANN model parameters including inputs are selected simultaneously and automatically. It reduced the computation time by eliminating trial and error approaches and provided better performance.

Comment 6: There is quite some improvements needed in the general writing and phrasing. Some sentences were hard to follow, and I think the manuscript would benefit from a general revision of the text. I have pointed out in the attached .pdf where I found the text could be improved. There is also a subtantial use of acronyms throughout the manuscript, which turn sometimes the reading difficult. As such, I would remove substantially the amount of acronyms used (e.g. variable names, Mahanadi river.

Response: Thank you for your valuable comments and suggestions. We have corrected it and updated in the revised manuscript.

Reviewer 2-Comments-answer (Pdf Comments)

Comment 1: What are the main conclusions of the study in terms of key variables to explain SSY?

Response: Thank you for your valuable comments.

The temporal data (water discharge, rainfall and temperature) and spatial (rock type, relief and catchment area) data are the major controlling factors of suspended sediment yield. It was found that water discharge and suspended sediment yield of Mahanadi river show wide spatial variation. The temperature and rainfall did not show the much variation among different gauge stations in the basin. Further, Tikarapara station has the highest water discharge, rainfall, catchment area and suspended sediment yield, whereas Andhiyakore station has the lowest suspended sediment yield. Suspended sediment yield was more non-linear complex processes as comparative other hydro-climatic variable (water discharge, rainfall and temperature). It was found that water discharge and rainfall are the most dominant controlling parameters of suspended sediment in the Mahanadi river. Effect of temperature in suspended sediment yield was found to be low. Temperature controls the sediment yield in several indirect ways. The change in temperature may affect the sediment discharge by altering runoff and changing the erosion rate through its influence on evapotranspiration, vegetation and weathering It was also concluded that suspended sediment was highly correlated to water discharge. The monsoon season accounts for most of the annual water discharge in the basin. The suspended sediment yield was more scattered than the water discharge, rainfall and temperature, and highly erratic in nature.

Comment 2: nonlinearity only follows the power law function.

Response: Thank you for your valuable comments.

The relationship between sediment yield (S) and water discharge (Q) is given by sediment rating curve (SRC) method as power law function or power relation which is given below

where a and b are the regression coefficient.

Comment 3: Q, R, T and SSY-Those have also a spatial dimension (how do they vary throughout the watershed).

Response: Thank you for your valuable comments. The Q, RF, T and SSY are the temporal data. It means that it varies time to time. These data are considered on the specific gauge locations. Different gauging stations have different values of Q, RF, T and SSY. The figure 2 represents the spatial variation of these variable at different gauging stations. It was found that water discharge and suspended sediment yield of Mahanadi river show wide spatial variation. The temperature and rainfall did not show the much variation among different gauge stations in the basin. Further, Tikarapara station has the highest water discharge, rainfall, and suspended sediment yield, whereas Andhiyakore station has the lowest suspended sediment yield.

Comment 4: Are the variables scaled with the maximum and mininum measured for each variable, 1 being the maximum and 0 the minimum?

Response: Thank you for your valuable comments. Yes, “0” is the minimum and “1” is the maximum value.

Comment 5: I would really combine Figure 2, 3, 4 and 5 into a nice map showing the spatial distribution of the different variables across the Mahanadi watershed.  Perhaps an histogram for each of the station? Table 2 can go in a Supporting Information since it is repetitive of the figures.

Response: Thank you for your valuable comments. We have combined the Figures 2, 3, 4 and 5. All these figures demonstrates about the annual variation of water discharge, rainfall, temperature and suspended sediment yield. Table 2 demonstrates about the monthly variation of water discharge, rainfall, temperature and suspended sediment yield. Thus ,Table 2 are different from the Figure 2, 3, 4 and 5. We will implement your other some valuable comments in future research.

Comment 6: Please specify what the different entries of the Table are:  Training, validation, testing : are these taking all the sites together? For each specific site: it the values the results of the validation subset, or testing subset?

Response: Thank you for your valuable comments. Yes, we have combined all eleven gauging stations data in the training, validation and testing data. Other results represent the testing results of each specific site.

Comment 7: Table-5-Not sure this table is needed. The difference in the peaks could be discussed from Figure 7.

Response: Thank you for your comments. Table 5 describes the values of rainfall, water discharge and temperature corresponding to peak suspended sediment yield of Figure 7 for well understanding. Figure 7 demonstrates the peak values of observed and predicted suspended sediment yield only. This Table represents the status of the controlling factors of SSY at the peaks time. This Table information supports for analysing and well explaining the peak SSY values. It also helps in the interpreting the peak situations of SSY of Figure 7.

Comment 8: Why are there so many months where observed SSY data are zero (or close to zero). Perhaps a log-log plot would help the visualization?

Response: Thank you for your comments. There are so many monthly SSY value is zero which may be in the non-monsoon season. There are some non rainy months. It may be summer season. In the summer season, SSY may be zero or close to zero.  In this study, we have used all monsoon and non-monsoon season data.

Reviewer 3 Report (New Reviewer)

            This manuscript brings a new toll known as artificial intelligence that is beginning to be used in water resources. This technology is powerful and very useful, and the manuscript is very well written and has scientific merit; however, the authors have to attend to two significant and mandatory aspects for this manuscript to be considered for publication:

First, they have to describe the hybrid artificial intelligence model with an artificial neural network (ANN) and the genetic algorithm (GA). I know that the “Water” asked for short manuscripts, but the minimum and intelligible model description is necessary,

The second one is how this automatic gauge works. How was the suspended sediment sampled? What known international methodology was followed in the sediment suspended sampled?

This is a major review; after that, I can consider this manuscript published.

 Add to title: Optimized scenario for estimating suspended sediment yield using an artificial neural network coupled with a Genetic algorithm,

Figures 2, 3, 4, 5, 6, 8, 9, and 10 have no good quality.

Author Response

Answers to Reviewers’ and Associate Editor Comments on Manuscript ID: water-1876557

We would like to thank all of the reviewers, associate editor, and handling editor. We have revised our manuscript according to their comments. In this file, we have answered all questions and comments the reviewers’ and associate editor have pointed out, and showed what changes we have made in the revised manuscript. All reviewers’ comments and questions are shown as black colour in this file, our answers to reviewers’ comments are show as RED colour, and the changes in the revised manuscript is shown in GREEN colour. We have addressed individual reviewers’ and associate editor’s comments separately.

Reviewer 3-Comments-answer

Comment 1: First, they have to describe the hybrid artificial intelligence model with an artificial neural network (ANN) and the genetic algorithm (GA). I know that the “Water” asked for short manuscripts, but the minimum and intelligible model description is necessary.

Response: Thank you for your valuable comments and suggestions. The following lines are added in the revised manuscript.

Numerous studies have been conducted recently in a variety of fields using hybridised genetic algorithm-based artificial neural networks, which are gaining popularity among researchers. These studies have shown that these hybrid approaches are capable of enhancing the system's accuracy. The hybrid artificial intelligence demonstrated the use of heuristic and meta-heuristic approaches for simultaneous optimization of associated parameters in artificial intelligence models to overcome the limitation of trial-and-error methods and overfitting and underfitting problems. Several research shows that these approaches are not only reduce the computational intensiveness, but also provide superior results.

Comment 2: The second one is how this automatic gauge works. How was the suspended sediment sampled? What known international methodology was followed in the sediment suspended sampled?

Response: Thank you for your valuable comments and suggestions.

The most common type of automatic water level recorder uses a float line with a metal float at one end and small counter weight at the other end. The float line passes over a pulley and transfers the changes of water level to it. A recording stylus is attached to the pulley. It moves laterally and traces the water level fluctuations on a recorder chart. The recorder chart is a tracing quality strip paper wound over rollers or a drum. The recorder chart is connected to a clockwork mechanism which moves it at pre-determined speed continuously

       Suspended sediment concentrations are typically measured by collecting samples of water-sediment mixtures. Bottle samples are the traditional method for obtaining suspended sediment samples and may be collected using either depth-integrated or point-integrated methods [Gray and Landers, 2014].

       Depth-integrated sampling involves lowering the sediment samples from the river surface to the bed of the channel at a uniform rate while a bottle within the sampler collects an incremental volume of the water-sediment mixture from all points along the sampled depth. Each location chosen for a measurement is known as a sampling vertical and the movement of the sampler from the surface to the bed, or vice versa, is known as a transit. Point-integrated sampling involves lowering the sampler to a specific depth in the water column and collecting a volume of water-sediment mixture at a particular point in the flow [Tassone and Lapointe, 1999].

References

Gray, J. R., and M. N. Landers (2014). Measuring suspended sediment, inComprehensive Water Quality and Purification, vol. 1, edited byS. Ahuja, pp. 157–204, Elsevier, Amsterdam.

Tassone, B., and F. Lapointe (1999). Suspended-sediment sampling, Hydrometric technician career development program, The Water Surv. Of Can., Ottawa, Ontario, Canada. [Available at http://publications.gc.ca/collections/collection_2014/ec/En56-247-1999-eng.pdf.]

Comment 3: Add to title: Optimized scenario for estimating suspended sediment yield using an artificial neural network coupled with a Genetic algorithm,

Response: Thank you for your valuable comments. We have corrected and updated it in the revised manuscript.

Comment 4: Figures 2, 3, 4, 5, 6, 8, 9, and 10 have no good quality.

Response: Thank you for your valuable comments and suggestions. We have changed these Figures and updated it in the revised manuscript.

Round 2

Reviewer 1 Report (New Reviewer)

The manuscript has been revised and improved according to the comments.

Author Response

Author would like to thank to the reviewer. 

Reviewer 2 Report (New Reviewer)

Dear Authors,

Thank you for the revisions you performed on your manuscript entitled:"Optimized scenario for estimating suspended sediment yield using an artificial neural network coupled with Genetic algorithm". 

I do find that the revised version of the manuscript improved the overall quality of the paper. Yet, not all the comments I had raised in the first round of revisions were adressed, and there are still elements of the revised version that could be improved in my view (comments in the attached .pdf file).

I summarize here the main elements from which your manuscript would in my view benefit, and/or I would appreciate to see:

(1) In general, in the Response-to-Reviewer file, every comments (and not only a selection) should be listed, with either the author response or rebuttal, and with the clear statement of what part of the text (lines) has been modified. In your response to the first review, you provided with interesting responses, but there are not always implemented into the main manuscript, which is actually the important part of the reviewing process. It is important that all comments of both the first and second reviews are clearly answered (with modified lines) or rebutted with appropriate argumenation.

(2) I still find that there is quite room for improvement in the way the main results are presented. Tables are hard to read, and I think the manuscript would benefit from more graphically-presented results. I still find that maps, instead of histograms, may be a nice way to present the main results to better visualize the spatial variability in the variables of interest.

(3) The paper would benefit from an editing of the English by a specialized service.

I hope the review will help the authors to improve their manuscript.

#End of review

Author Response

Reviewer 2-Comments-answer

Comment 1: In general, in the Response-to-Reviewer file, every comments (and not only a selection) should be listed, with either the author response or rebuttal, and with the clear statement of what part of the text (lines) has been modified. In your response to the first review, you provided with interesting responses, but there are not always implemented into the main manuscript, which is actually the important part of the reviewing process. It is important that all comments of both the first and second reviews are clearly answered (with modified lines) or rebutted with appropriate argumenation.

Response: Thank you for your valuable comments. Now, we have addressed the all comments as per your valuable suggestions. Now, it is corrected in the revised manuscript.

Comment 2: I still find that there is quite room for improvement in the way the main results are presented. Tables are hard to read, and I think the manuscript would benefit from more graphically-presented results. I still find that maps, instead of histograms, may be a nice way to present the main results to better visualize the spatial variability in the variables of interest.

Response: Thank your valuable comments and suggestions. Your given suggestions are very nice but if we plot all histogram plots of four temporal variables like water discharge, rainfall, temperature and suspended sediment yield for eleven gauge stations then diagram is getting very complex. Each gauge station contains the four histogram plots of water discharge, rainfall, temperature and suspended sediment yield. So total number of plots for all eleven stations will be 44 in single plot which may increase the manuscript size also. Moreover, all data sets have wide range. These data have different dimensions and different units. If plot in single map or histogram plot, then all data points can not be visible clearly due to wide range of different data sets. Temperature data has range in hundred, but sediment data has range in the millions. There are large of number of approaches of plotting the data are available in the literature; however, these types of plots are widely used in the sediment yield modelling community for representing the variation of data sets (Bastia and Equeenuddin 2016, Yadav et al. 2017). Therefore, we have decided to use these types of approaches for plotting the data for our research. We will implement the your valuable idea for future research. We have plotted the histogram plots of water discharge, rainfall, temperature and suspended sediment yield using combined data of all eleven gauging stations and removed the Table 2 from the revised manuscript.

References

  1. Bastia, F.; Equeenuddin, S.M., Spatio-temporal variation of water flow and sediment discharge in the Mahanadi River, India. Global and Planetary Change, 2016, 144, 51-66.
  2. Yadav, A.; Chatterjee, S.; Equeenuddin, S.M., Prediction of Suspended Sediment Yield by Artificial Neural Network and Traditional Mathematical Model in Mahanadi River Basin, India, Journal of Sustainable Water Resource Management, 2017, 4(4), 745-759.

The following lines are added in the revised manuscript.

The histogram plots of monthly hydro-climatic (Q, R, T and SSY) data is presented in Figure 2. It is observed that the water discharge, rainfall, and suspended sediment yield are positively skewed (right asymmetry), while the temperature is negatively skewed (left asymmetry). Suspended sediment yield has a higher skew than other variables. High skewness values have a negative impact on performance of the ANN model[99].

Figure 2. Histogram plots of hydro-climatic data in MR basin (a) Water discharge (b) Rainfall (c) Temperature (d) Suspended sediment yield

Comment 3: The paper would benefit from an editing of the English by a specialized service.

Response: Thank you for your valuable suggestions. Finally, English correction is done and grammatical errors are removed in the revised manuscript.

Pdf. Comments-Response (Reviewer 2)

Comment 1: Lines 22-23-Construction of water resources means nothing to me. The sentence should be rephrased to target appropriate challenges.

Response: Thank you for your valuable comments. Now, it is corrected and highlighted by GREEN color in the revised manuscript (Lines: 22-23).

Comment 2: Lines: 32-34, non-capital letter

Response: Thank you for your valuable comments. Now, it is corrected in the revised manuscript and highlighted by GREEN color (Lines:32-34).

Comment 3: Lines-34, missing space

Response: Thank you for your valuable comments. We have corrected and highlighted it by GREEN color in the revised manuscript (Line:34).

Comment 4: Lines-36, provided with

Response: Thank you for your valuable comments. As per your comments, now, “provided the” is replaced by “provided with”. We have corrected and highlighted it by GREEN color in the revised manuscript (Line: 37).

Comment 5: Line-39, I would group this sentence with the following one:"most appropriate model compared to other examined models for estimating SSY in the MR basin, India, in particular at the Tikarapara measuring station.

Response: Thank you for your valuable comments. We have corrected and highlighted it by GREEN color in the revised manuscript (Lines:39-40).

Comment 6: Lines:60-61, I still do not get this bit.

Response: Thank you for your valuable comments. We have corrected and highlighted it by GREEN color in the revised manuscript (Line: 57). Now, it is updated as “deposition of SSY diminishes the water storage capacity”.

Comment 7: Lines: 65-67, In addition, various researchers revealed that the impact of the sediment on the erosion of hydro-turbine components in hydro power plant [7, 12-14]. Poor wording.

Response: Thank you for your valuable comments. Now, we have corrected and highlighted it by GREEN color in the revised manuscript (Lines: 62-64).

Comment 8: Lines:68-69, at a particular time assists water resource system managers and planners in better understanding the system's issues and finding alternative solutions to these issues [15].

Response: Thank you for your valuable comments. Now, we have corrected and highlighted it by GREEN color in the revised manuscript (Lines: 65-67).

Comment 9: Lines: 81-82, The majority of the water discharge in the river is caused by precipitation. Back up this statement with references. How about the role of groundwater with respects to rainfall?

Response: Thank you for your valuable comments. Now, we have corrected and highlighted it by GREEN color in the revised manuscript (Lines: 81-88). The following lines are added ain the revised manuscript.

As a peninsular river, the majority of the water discharge in the Mahanadi river is contributed by precipitation during monsoon season and groundwater recharge accounts for small contribution [29,31]. Rainfall is an important factor which affecting in many ways the groundwater resources in an area and there is also considerable flow of water in some locations during non-monsoon which may be due to additions from groundwater to the river during non-monsoon periods [29,31].

Comment 10: Line 85, which causes.It should be update by"where steep catchments are associated with high rate of erosion and sediment load".

Response: Thank you for your valuable comments. Now, we have corrected and highlighted it by GREEN color in the revised manuscript (Lines: 93-94).

Comment 11: Line: 88, “is” should be replaced with “is also”

Response: Thank you for your valuable comments. Now, we have corrected and highlighted it by GREEN color in the revised manuscript (Lines: 97).

Comment 12: Line 89, significant factors of sediment like. Poor wording.

Response: Thank you for your valuable comments. Now, we have corrected and highlighted it by GREEN color in the revised manuscript (Lines: 97-101).

Comment 13: Line:90, Be consistent: either capital letters to every variables, or non capital letters.

Response: Thank you for your valuable comments. Now, we have corrected and highlighted it by GREEN color in the revised manuscript (Lines: 98-99).

Comment 14: Line 92, on the basis of previous research. Add some references.

Response: Thank you for your valuable comments. Now, we have corrected and highlighted it by GREEN color in the revised manuscript (Line: 101).

Comment 15: Line: 94, “by some researchers” should be replace with “in previous research”

Response: Thank you for your valuable comments. Now, we have corrected and highlighted it by GREEN color in the revised manuscript (Line: 109).

Comment 16: Line 100, replace the other by multiple.

Response: Thank you for your valuable comments. Now, we have corrected and highlighted it by GREEN color in the revised manuscript (Line: 114).

Comment 17: Line 100, I would list all the most important, not only two.

Response: Thank you for your valuable comments. Now, we have corrected and highlighted it by GREEN color in the revised manuscript (Line: 115).

Comment 18: Line 103, “which is widely” should be replace by “and is now widely used”.

Response: Thank you for your valuable comments. Now, we have corrected and highlighted it by GREEN color in the revised manuscript (Line: 118).

Comment 19: Line:106, widely should be replace by Notably.

Response: Thank you for your valuable comments. Now, we have updated the “notably” instead of “widely” in the revised manuscript which is highlighted by GREEN color (Line: 124).

Comment 20: Line 293, “are” should be replace with “were”.

Response: Thank you for your valuable comments. Now, we have corrected and highlighted it by GREEN color in the revised manuscript (Line: 154).

Comment 21: Lines: 301-302, The GA and ANN approaches are successfully applied for predicting sediment in river basin system [40,74-77]. It should be (...) and ANN-GA approaches were successfully applied in other settings for predicting sediment in river basin systems.

Response: Thank you for your valuable comments. Now, we have corrected and highlighted it by GREEN color in the revised manuscript (Lines: 166-168).

Comment 22: Lines: 304-310, The way the argument is built is not the best in my view. I would state what has been done in previous research to predict the SSY, what does not work or is missing, and how the ANN-GA approach could help.

Response: Thank you for your valuable comments. Now, we have corrected and highlighted it by GREEN color in the revised manuscript (Lines: 172-184).

Comment 23: Lines: 374-375, there is no any such the single objective ANN-GA. poor wording.

Response: Thank you for your valuable comments. Now, we have corrected and highlighted it by GREEN color in the revised manuscript (Lines: 203-204).

Comment 24: Line: 390, “Mahanadi river” should be replace with “The Mahanadi river”.

Response: Thank you for your valuable comments. Now, we have corrected and highlighted it by GREEN color in the revised manuscript (Line: 220).

Comment 25: Line 394-395, The catchment area contribution of the river is 53%, 46% in Chhattisgarh and Odisha with remaining in Maharashtra and Jharkhand, respectively.

Response: Thank you for your valuable comments. Now, we have corrected and highlighted it by GREEN color in the revised manuscript (Lines: 225-227). The following lines are added in the revised manuscript.

The catchment area contribution of the river is 53% (75136 km2) in Chhattisgarh, 46% (65580 km2) in Odisha with remainder of the basin is in the Maharashtra and Jharkhand states [84].

Comment 26: Line 396, “makes up” should be replace with “cover”.

Response: Thank you for your valuable comments. Now, we have corrected and highlighted it by GREEN color in the revised manuscript (Line: 230).

Comment 27: Lines: 400-401, generously proportioned

Response: Thank you for your valuable comments. Now, we have corrected and highlighted it by GREEN color in the revised manuscript (Line: 236).

Comment 28: Line: 405, downstream in one word

Response: Thank you for your valuable comments. Now, we have corrected and highlighted it by GREEN color in the revised manuscript (Line: 241).

Comment 29:  Lines: 412-413, The maximumt and lowest relative humidity range is 68 to 87% and 9 to 45%, respectively. The average highest and minimum relative humidity of the basin is 82% and 31.6% respectively [87]. I don't think this is needed with respects to the study.

Response: Thank you for your valuable comments. As per your valuable comments, Now, we have deleted it form the revised manuscript.

Comment 30: Line: 468, 34% should be update.

Response: Thank you for your valuable comments. Now, we have corrected and highlighted it by GREEN color in the revised manuscript (Lines: 259-262).

Comment 31: Line: 658, “meeting” should be replace with “confluence”.

Response: Thank you for your valuable comments. Now, we have corrected and highlighted it by GREEN color in the revised manuscript (Line: 504).

Comment 32: Line 671, p should be capital in “Pearson”.

Response: Thank you for your valuable comments. Now, we have corrected and highlighted it by GREEN color in the revised manuscript (Lines: 517, 521).

Comment 33: Line: 889, geo morphological should be in one word.

Response: Thank you for your valuable comments. Now, we have corrected and highlighted it by GREEN color in the revised manuscript (Lines: 682-683).

Comment 34: Line: 916, Recent time, in hydrological research and other disciplines. Poor wording.

Response: Thank you for your valuable comments. Now, we have corrected and highlighted it by GREEN color in the revised manuscript (Lines: 710-711).

Comment 35: Line: 1087, “ranfall” should be update with “”rainfall.

Response: Thank you for your valuable comments. Now, we have corrected and highlighted it by GREEN color in the revised manuscript (Line: 788).

Reviewer 3 Report (New Reviewer)

Dear authors 

Please, include in the material and methods you rely upon how the automatic gage works to sample suspended sediment and flow data.

Author Response

Reviewer 3-Comments-answer

Comment 1: Please, include in the material and methods you rely upon how the automatic gage works to sample suspended sediment and flow data.

Response: Thank you for your valuable comments. The following lines are added in the revised manuscript. All the data of this study are collected from the Central Water Commission (CWC), Bhubaneswar, Odisha, India. The following lines are added in the revised manuscript(Lines: 378-392).

The most common type of automatic water level recorder uses a float line with a metal float at one end and small counter weight at the other end. The float line passes over a pulley and transfers the changes of water level to it. A recording stylus is attached to the pulley. It moves laterally and traces the water level fluctuations on a recorder chart. The recorder chart is a tracing quality strip paper wound over rollers or a drum. The recorder chart is connected to a clockwork mechanism which moves it at pre-determined speed continuously. Suspended sediment concentrations are typically measured by collecting samples of water-sediment mixtures. Bottle samples are the traditional method for obtaining suspended sediment samples and may be collected using either depth-integrated or point-integrated methods [97,98]. Depth-integrated sampling method is generaly used which involves lowering the sediment samples from the river surface to the bed of the channel at a uniform rate while a bottle within the sampler collects an incremental volume of the water-sediment mixture from all points along the sampled depth. Each location chosen for a measurement is known as a sampling vertical and the movement of the sampler from the surface to the bed, or vice versa, is known as a transit.

This manuscript is a resubmission of an earlier submission. The following is a list of the peer review reports and author responses from that submission.

Round 1

Reviewer 1 Report

Review of manuscript “Optimized scenario for estimating suspended sediment yield using an artificial neural network coupled with Genetic algorithm”, by Yadav et al., submitted to WATER

30 June 2022

The manuscript presents the determination of suspended sediment yield for a large river basin in India. The determination was performed using jointly ANN and GA, and this hybridization returned, according to the authors’ claim, a significant improvement in prediction performance compared to alternative methods.

I have to admit that at a certain point I had to surrender, because the text is quite difficult to follow and many detailed aspects of the computation procedure are beyond my understanding. However, I would raise few major points.

First, the novelty of the study is unclear. There are frequent statements that this is the first time that such estimation was done for this catchments, but this is not enough for scientific publication. There are also some statements pointing at a methodological advance, but they are not given much relevance.

Second, my eye fell on Figure 9 that does not really show a large difference between the performances of the alternative methods. Can we really say that this GA-ANN is much better than others?

Third, the text is in a very bad shape and in huge need of polishing/clarification.

Some comments follow (until, as mentioned, I stopped).

42-43: for this to be true, one should demonstrate that the model parameterization is limitedly site-specific. Since only one basin was considered in this study, such a demonstration seems still to come.

92-115: this paragraph is too loopy and highly needs polishing. For example, the problem of under- and over-fitting is mentioned twice (at 92 and 107) while repetitions need to be avoided. Same for using jointly ANN and GA (97 and 115).

127: please repeat the full name of the river in the main text as the acronym MR was defined only in the abstract.

131: the authors should clarify why the application of the model to the entire basin rather than to sub-basins is a challenge and, in turn, an aspect of novelty of the present study. Is it a problem of parameter scaling, or of result composition? Maybe something in this respect will come in the following.

146: similarly, the fact that an estimate of the SSY for this basin previously did not exist is an important technical result, but does not justify publication in a scientific journal. Is there a significant methodological advance?

168: Q is normally used for flow rates that are not in mm. Please clarify.

284: weird units for Q are used here.

181-185: this would be better placed under “methodology”, I think.

211: this sounds like novelty, should be given more relevance in the introduction. I skip the rest of section 2, that is beyond my understanding.

328-341: this is an example of text that makes a reader stop. Which is the need to just repeat values that are depicted in a plot?

356: the story is becoming extremely obscure. It sounds like I am unable to usefully review this manuscript. Sorry but I quit.

Reviewer 2 Report

1-      It is suggested that the first paragraph of the introduction regarding the effects of sedimentation and flooding, which is not very related to the topic of the manuscript, should be rewritten to be summarized and focused on the topic of the research.

2-      In the second paragraph of the introduction, there are many short sentences that are recommended to be rewritten. Also, there are unrelated sentences in the second paragraph, for example, “Climate change will affect water resources and hydrological processes “

3-      relief(R), and catchment area (CA) are not geological variables.

4-      There is no logical relationship between different paragraphs of the introduction, and repeated sentences are observed in the introduction, including the limitations of the artificial intelligence method.

5-      I do not agree with this sentence “Methods for dealing with data with non-stationarity data have not been well suitable for hydrological forecasting and prediction problems”. It is required more clarification for this issue

6-      1- It is not acceptable to refer to the MR River as a research area, and the ability to generalize the results to other rivers should be considered. In general, according to the authors' reference to previous research in the field of GA-ANN in the field of bed load which gas more complexity and nonlinearity in comparison to SSY, the novelty of the manuscript is not very significant.

7-      According to this sentence “The GA-ANN models have fruitfully been applied from prediction as well as forecasting perspective of stream flow, flood, bed load transport, Q and run-off” the novelty of the manuscript is not understood

8-      mention the research results at the end of the introduction is not common

9-      On page 4, line 168, precipitation is usually expressed in millimeters, and it is not usual to express flow discharge in millimeters “During the study, it has found that the average annual Q varies from 1200-1400 mm 168 in the year”

10-   On page 5, line 182, what is the meaning of MOO, which is mentioned for the first time?

11-   This sentence on page 5, line 190 is not correct “Data are divided into testing (15 percent), validation (15% percent) and testing(15 percent) [25,36,84]

12-   On page 6, line 268, what does WD mean?

13-   Spatial variation of data should be moved in data and methodology. It is not part of your results

14-   Page 9 line 329 there is a dictation error

15-   In order to select the inputs of the neural network model, sensitivity analysis is definitely necessary, while the use of a daily time scale for the model inputs will definitely lead to better results in the predictions. Also, in this case, the use of delays of a few days before for prediction SSY can aim for more accurate results. This issue should be discussed by the authors

16-   Negative values estimation by GA-ANN model are not acceptable and it can be reasonable while the values of SSY for training have positive values. It shows the proposed model cannot work satisfactory

17-   In Fig.8, you must use R2 (determination coefficient) instead of correlation coefficient to show the performance of the model

18-   I could not understand the difference between Q(m3/s) and Q(mm) in table 3. It is not common to use a variable with two units at all.

19-   The author’s discussion according to table 3 is not strong enough. It is suggested to show the correlation between discharge and SSY at different stations then you can have a discussion between the Peak of hydrograph for discharge and SSY

20-   Again I could not understand your meaning for GA-MOO-ANN at line 461

21-   According to Fig.9, I could not find any superiority for the proposed model GA-ANN regarding ANN and MLR models. Again I could not understand using R instead of R2 for comparison

Reviewer 3 Report

- The paper in overall has presented three model comprising ANN, ANN-GA and MLR as well as sediment rating curve, where the final conclusions are totally expectable. This means that metaheuristic algorithms definitely can increase the accuracy while MLR due to inherent limitation cannot be competitive. In this point of view the paper doesn’t have any novelty. On the other hand, the authors stated that in the absence of gauging data the model can be used, this is very obvious target of predictive model which is carried out through the weight database and thus never ever can be considered as novelty. According to these descriptions, the current paper doesn’t meet any novelty and significant of contribution criteria. 

- The main drawback and wondering point in this paper is the selective randomized data (L191-197)????? This is not randomization as you just selected those you want. You are not allowed at all to use such randomization except if you consider the time dependent or time series data. As this work doesn’t fall in time series data, this randomization conceptually and clearly has great conflicting with randomization theory in AI.  

- The next issue refers to applied model and adjusted internal hyper parameters. Nothing, I emphasize, nothing concerning the modeling can be found. How the optimum topology for ANN has been achieved? With what criteria you found that it can be the most appropriate candidate? How the internal hyper parameters have been adjusted? With what training strategy? How the number of layers, number of neurons have been managed? What about the activation functions? Learning rate? How the overfitting problem, vanishing, gradient exploding, early convergence and getting stuck into local minima has been cured? How the error improvement in each epoch was monitored? How did you become assure that the activation function hasn’t been saturated? What about the termination criterion (criteria)? If it is not achieved, then what will happen?? How the hyperparameters of GA have been considered? what about the mutation rate? Anything concerning the cross over? Nothing about the generations???? Totally unacceptable and incredible.

- This paper presents the results in such way that all data are accurate and there is no uncertainty in the used data while the predicted results clearly show great uncertainty. I curiously would like to see how the uncertainty involved in the datasets and predicted values have been evaluated? What about the reliability-based analysis? Urgently looking at https://link.springer.com/article/10.1007/s11053-022-10051-w is mandatory.

The remain technical concerns are as follows: 

1. First of all, the English of the work significantly suffers from frequent linguistic flaws, word repetitions, punctuation problems as well as long and vague statements. Must and have to be revised by native expert or verified proofread systems. Simply can be seen from the Abstract. Just some other random samples: L154 Sq.km or Km2??? L155-157, contribution of what??? L157 after ‘.’ With small letter?...

2. The paper has been submitted to a specified journal in the field of water and resource engineering and the readers also know about concept of river engineering, water resource and corresponding management problems. In this point of view, the Abstract suffers from the repetition of known materials which definitely must be truncated. It is well recognized that Abstract should be short, concise but self-informative and to the point. Here is far from such characteristics. From L21-27 denote totally obvious statements. In another case, randomization is an obvious and mandatory operation and you cited it to 3 scholars (L191)?????? As all the scholars used the randomization you must cite it to all worldwide scholars why just these three?????  L476-478, again you have citations to show ANN is better than MLR????? 

3. SSY???? Do you mean for suspended sediment load??? If yes, why did you change the known and accepted terms??? If not, what does exactly mean? What is the different between SSY and SSL???

4. Keywords should be representative and available in both Abstract and context. Genetic algorithm and artificial neural network are general terms and doesn’t show any specificity in this work.

5. This paper suffers from a huge amount of used abbreviations which makes it very boring to follow. 

6. The introduction in terms of both technically and literature review must be updated. Technically, giving a bunch of references cannot correspond to depth of review. For example, L65-66, it is widely recognized and just for an obvious statement you gave 7 references!!!! Form the whole introduction, there is no possibility for quick catch on what the problem is what you are pursuing? What is the gap of previous models and which part their limitation is going to be filled here? With what method and why? what motives for? What is the main novelty of applied method? The main advantage of the used method rather than previous models? Significant of contributions? In the case of literature review, many papers dealing with novel AI methods like https://www.tandfonline.com/doi/abs/10.1080/02626667.2021.2003367, https://www.frontiersin.org/articles/10.3389/fenvs.2022.821079/full, https://link.springer.com/article/10.1007/s11356-020-11335-5, https://link.springer.com/article/10.1007/s40808-021-01165-w, ... Must be added. In the field of effect of rock or subsurface distributed materials on sediment loads https://www.sciencedirect.com/science/article/abs/pii/0022169483900823, https://www.sciencedirect.com/science/article/abs/pii/S0167880920303005, https://www.sciencedirect.com/science/article/abs/pii/S0169555X12000815, https://www.sciencedirect.com/science/article/pii/S0341816222002752, and … must be considered.

7. The last paragraph of the Introduction should be assigned to brief summary of applied method and bolded findings. 

8. You have used the GA. Why? why didn’t use other metaheuristics? Why GA has been selected? according to which advantage?

9. Why the quality of figures and texts are different?

10. In Fig7 and 8, you have negative predicted SSY?????? What does negative values mean? Physical interpretation???

11. Wondering for long introduction on study area. We are not here to read the geography of the area or what stuffs are within. You must clearly introduce where the study area is and how many data with what characteristics for which time interval from how many stations with what distances have been gathered. How the data has been processed and screened? Must be given in a table with simple statistical descriptions. You have used many abbreviations which make it very hard to follow. I totally was lost with normalization. 

12. Could you please let the reader know if you wrote the code? With what programming language? How are the weights saved? With what command it can be recalled? Given results only uncovers the samples and won't determine what variables have the most influence. Extraneous variables might interfere with the information and thus outcomes can be adversely impacted by the quality of the work. You must carry out the sensitivity analyses through the weight database for model calibration and variable reduction. Look at https://iwaponline.com/jh/article/22/3/562/72506/Updating-the-neural-network-sediment-load-models, …

13. In MAE, the gradient magnitude is not dependent on the error size, only on the sign of y – Å· and thus the gradient magnitude will be large even when the error is small, which in turn can lead to convergence problems. How did you overcome on this issue??

14. Where the error has been considered? Mathematically, considered VAR gives added weight to outliers, where is you interpretation and treatment????

15. The root of MSE should give the RMSE. Look at Table 2 and clearly approve that howe for example MSE 2.39x10-5 can give RMSE 0.006 which must be 0.0048???? or 2x10-4 should give 0.014 why 0.012????... 

I don’t go through the detail and to save the time gave the rest up.